# Multiple roles for hypoxia inducible factor 1-alpha in airway epithelial cells during mucormycosis

Povilas Kavaliauskas[1,7], Yiyou Gu[2,7], Naushaba Hasin[3,6,7], Karen T. Graf[3], Abdullah Alqarihi [2], Amol C. Shetty [3], Carrie McCracken[3], Thomas J. Walsh[1,4], Ashraf S. Ibrahim [2,5] & Vincent M. Bruno [1,3] ✉

During pulmonary mucormycosis, inhaled sporangiospores adhere to, germinate, and invade airway epithelial cells to establish infection. We provide evidence that HIF1α plays dual roles in airway epithelial cells during Mucorales infection. We observed an increase in HIF1α protein accumulation and increased expression of many known HIF1α-responsive genes during in vitro infection, indicating that HIF1α signaling is activated by Mucorales infection. Inhibition of HIF1α signaling led to a substantial decrease in the ability of *R. delemar* to invade cultured airway epithelial cells. Transcriptome analysis revealed that *R. delemar* infection induces the expression of many pro-inflammatory genes whose expression was significantly reduced by HIF1α inhibition. Importantly, pharmacological inhibition of HIF1α increased survival in a mouse model of pulmonary mucormycosis without reducing fungal burden. These results suggest that HIF1α plays two opposing roles during mucormycosis: one that facilitates the ability of Mucorales to invade the host cells and one that facilitates the ability of the host to mount an innate immune response.

Mucormycoses are life-threatening invasive infections caused by filamentous fungi of the order Mucorales. There are twenty seven different species of Mucorales that have been identified as causative agents of mucormycosis[1]. The most commonly isolated organisms from patients belong to the genus *Rhizopus* (including *R. delemar* and *R. oryzae)* and account for approximately 70% of all cases[2–4]. *Mucor* spp., *Lichtheimia* spp., and *Cunninghamella* spp. are also significant causes of infections and account for 45% of European cases, combined[5].

There are few antifungal agents that have been approved by the United States FDA for treating mucormycosis; the first was amphotericin B (AmB) which has significant nephrotoxicity and very limited

clinical success[3]. Lipid formulations of AmB are less toxic and are considered as first-line therapy[6]. However, patients may be refractory to or intolerant of these drugs[7]. Isavuconazole is also approved to treat mucormycosis as first-line therapy and posaconazole is used in salvage therapy; however, neither of these triazoles is superior to AmB[6]. In addition to antifungal therapy, the current standard of care for treating mucormycosis includes surgical debridement of the infected area and reversal of predisposing factors, including neutropenia, immunosuppression, and diabetes mellitus[7]. In the absence of surgical removal of the infected focus and reversal of predisposing factors, antifungal therapy alone is seldomly curative. Even when surgical debridement is combined with lipid formulation of AmB, the overall mortality

[1]Department of Microbiology and Immunology, University of Maryland School of Medicine, Baltimore, MD 21201, USA. [2]Division of Infectious Diseases, The Lundquist Institute for Biomedical Innovation, Harbor-UCLA Medical Center, Torrance, CA 90502, USA. [3]Institute for Genome Sciences, University of Maryland School of Medicine, Baltimore, MD 21201, USA. [4]Center for Innovative Therapeutics and Diagnostics, 6641 West Broad St., Room 100, Richmond, VA 23220, USA. [5]David Geffen School of Medicine at UCLA, Los Angeles, CA 90502, USA. [6]Present address: Millipore Sigma, 9900 Blackwell Road, Rockville, MD 20850, USA. [7]These authors contributed equally: Povilas Kavaliauskas, Yiyou Gu, Naushaba Hasin. ✉e-mail: vbruno@som.umaryland.edu

associated with mucormycosis exceeds 50% and can reach 90-100% in certain patient populations including those with persistent neutropenia, central nervous system infection, or disseminated disease[8–10].

Infections most often occur in individuals with predisposing risk factors that include neutropenia, hyperglycemia, and diabetic ketoacidosis (DKA), corticosteroid treatment, and hematological malignancies[4,11]. Furthermore, immunocompetent victims of natural disasters, including tornados, tsunamis and earthquakes, and traumatic events, such as burns and military combat injuries also are vulnerable to musculoskeletal mucormycosis[12]. Recently, it has been found that mucormycosis is also a significant complication of patients with severe infections caused by COVID-19[13–15].

The two most common forms of mucormycosis are sino-orbital/cerebral and pulmonary infections. Pulmonary disease is mainly found in patients rendered neutropenic by chemotherapy or in other immunocompromised hosts, particularly hematopoietic cell and solid organ transplant recipients. By comparison, sino-orbital/cerebral mucormycosis occurs predominantly in patients with diabetes mellitus. Disease caused by mucormycosis advances quickly and may cause extensive tissue damage by the time infection is diagnosed[16].

Airway epithelial cells line the lung mucosa and serve three important functions in the context of fungal respiratory infections: (i) they act as a physical barrier defense against inhaled fungal particles; (ii) they recruit innate immune cells to clear the infection via their ability to produce cytokines and chemokines[17]; and (iii) they produce and secrete antimicrobial peptides that can serve as a biochemical barrier to fungal invasion[18,19]. Many fungi, including Mucorales, have the ability to breach the physical epithelial barrier by inducing the host cell to endocytose the fungal spores and germlings[20–23]. Previous studies of the interaction between Mucorales and lung epithelial cells were performed using type II-like alveolar epithelial cells (AECs) A549 cell line which is derived from a lung tumor. In these studies, we showed that Mucorales are able to invade A549 alveolar epithelial cells and that this step is critical for disease progression[21,24–26]. Pharmacological treatments that inhibit Mucorales invasion of AECs in vitro also result in attenuation of virulence in a murine model of pulmonary mucormycosis[21,26]. Invasion is mediated by the action of the Mucorales encoded spore coat protein, CotH7, which binds to Integrin α3β1 expressed on the surface of the alveolar epithelial cells (AECs). This association triggers signaling of the epidermal growth factor receptor (EGFR) pathway and the subsequent endocytosis of the fungus[26].

Herein, we characterize the interaction between *R. delemar* and a normal small human airway epithelial cell line which was Tert-immortalized (HSAEC1-KT). We show that *R. delemar* can invade and damage cultured HSAEC1-KT cells and upregulate the expression of several pro-inflammatory genes that function in innate immunity. We also provide evidence that the hypoxia inducible factor 1-alpha (HIF1α) signaling pathway is activated following in vitro infection with *R. delemar* and other Mucorales. Inhibition of HIF1α accumulation by siRNA knockdown or with a small molecule inhibitor significantly reduced host cell invasion and prevented the infection-induced expression of genes encoding pro-inflammatory proteins including cytokines and chemokines. These results suggest that HIF1α facilitates multiple facets of the interaction between Mucorales and airway epithelial cells. Importantly, pharmacological inhibition of HIF1α, increased survival in a murine model of pulmonary mucormycosis.

## Results

### Interaction between *R. delemar* and HSAEC1-KT cells
We first tested the ability of *R. delemar* to invade and damage normal human small airway epithelial cells (HSAEC1-KT) during in vitro infection. Approximately 20% of the germlings had invaded a host cell by 3 h post-infection, and by 6 h post-infection, ~50% of the germlings had invaded a host cell (Fig. 1A). Fungal germlings damaged ~30% of the small airway epithelial cells by 48 h post-infection with no detectable

damage (*p*-value < 0.05 compared to 6-h timepoint) occurring until 12 h after addition of the germlings (Fig. 1B). These results demonstrate that *R. delemar* is able to invade and damage small airway epithelial cells and that fungal invasion precedes damage, a trend that matches that which has been previously reported for in vitro infections of both alveolar epithelial cells and nasal epithelial cells with *R. delemar*[26].

### RNA-seq analysis of in vitro *R. delemar* infection
We performed RNA-seq analysis on poly(A)-enriched RNA isolated from monolayers of human small airway epithelial cells infected with *R. delemar* for 3, 6 or 16 h. Our analysis also included uninfected samples that serve as negative controls. Each of the 4 experimental conditions were examined in triplicate. Since both fungi and humans are eukaryotes that produce poly-adenylated messages, the RNA preparations potentially contain a mixture of mRNAs expressed by *R. delemar* as well as by the host cells. However, since we did not include a bead-beating step in the RNA isolation protocol, we did not efficiently isolate fungal RNA from the infections. Therefore, we only examined host gene expression. From each of the 12 sequencing libraries, we obtained an average of $149.3 \pm 17.2$ million reads that mapped to the human reference genome (Supplementary Data 1). Analysis of human gene expression revealed 7776 genes that were differentially expressed (FDR < 0.01; $\log_2$ fold-change ≥ |1.0|) during at least one infection time-point compared to the uninfected control group. The complete set of differentially expressed genes can be found in Supplementary Data 2, 3 and 4.

### *R. delemar* induces innate immune responses in cultured small airway epithelial cells
The transcriptional program stimulated by *R. delemar* infection included the increased expression of several genes whose products are pro-inflammatory such as several chemokines, cytokines, and enzymes including CXCL1, CLCF1, CXCL10, CXCL11, CXCL16, CCL20, IL1α, IL1β, IL6, IL8, IL18, CSF2, CSF3, LIF, PTGS2, and GM-CSF (Fig. 1C, Supplementary Data 2, 3, 4). These transcriptomic results are supported by the infection-induced secretion of IL1α, IL1β, IL8, and GM-CSF (Fig. 1D).

### Upstream Regulatory Analysis of host gene expression
Using the Ingenuity Pathway Analysis (IPA) software (Materials and Methods), we performed an upstream regulator analysis (URA) on the sets of host genes that were differentially expressed (LFC ≥|1.0|, FDR < 0.01). Overall, our URA analysis predicted the modulation of 333 signaling proteins, of which 222 are predicted to be activated (Z-score ≥ 2.0) and 111 are predicted to be repressed (Z-score ≤ −2.0) (Supplementary Data 5).

Our analysis predicted the activation of several transcription factors (i.e. FOS, FOSL1, JUN, JUNB, EGR1, NFkB, and STAT3) and signaling pathways (i.e. ERK1/2 TLR3, TLR4, and TLR7/8) that are known to amplify innate immune responses (Fig. 2A, Supplementary Data 5). These results provide further evidence that *R. delemar* stimulates a pro-inflammatory transcriptional response in cultured airway epithelial cells that resembles the responses triggered by agonists of innate immune receptors or by other pathogens[27].

The predicted activation of a regulatory protein in our analysis may be indicative of an increase in activity resulting from a post-translational modification (e.g. phosphorylation), altered access to a cofactor, or a simple change in protein abundance. When the ERK signaling pathway is activated, threonine and tyrosine residues on ERK1/2 become phosphorylated and the signal is transmitted to a variety of downstream proteins[28]. *R. delemar* infection of the HSAEC1-KT cell line stimulated the phosphorylation of ERK1/2 when examined 1 h and 3 h post-infection by immunoblotting (Fig. 2B, C). This result provides an orthogonal validation of our URA analysis for a pathway that has not been previously tied to mucormycosis.

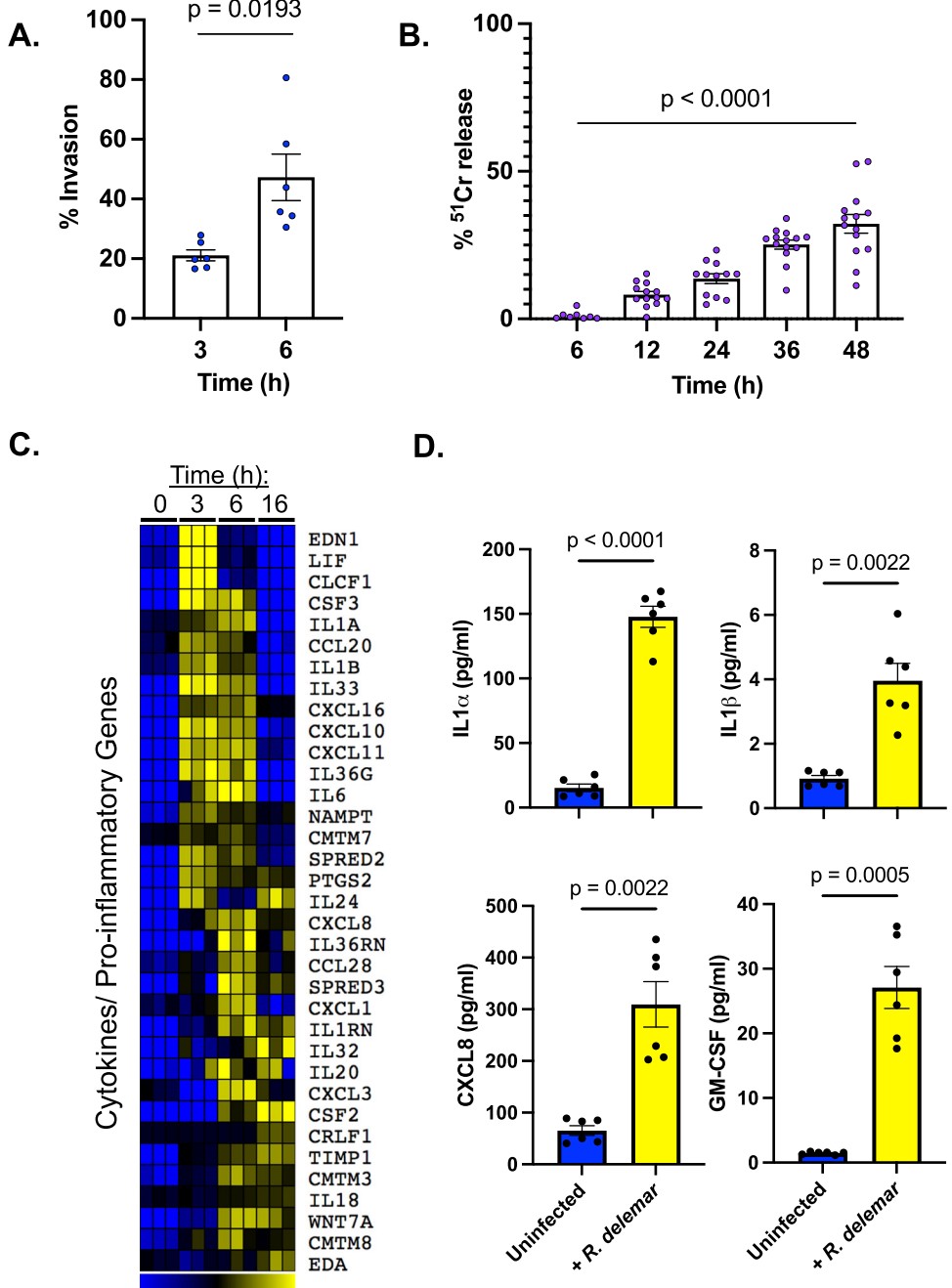

**Fig. 1 | Interaction of *R. delemar* with small airway epithelial cells. A** *R. delemar* invasion of HSAEC1-KT cells was determined with a differential fluorescence assay using 1% Uvitex. Data represents the mean ± SEM of two independent experiments performed in triplicate. *n* = 6. Two-tailed Student's *t* test was used for statistical analysis. **B** *R. delemar*-induced damage of HSAEC1-KT cell was determined with a 51Cr release assay. Data represents the mean ± SEM of two independent experiments with several biological replicates per experiment. *n* = 8 for 6 h, *n* = 13 for 12 h, *n* = 12 for 24 h, *n* = 14 for 36 h, *n* = 14 for 48 h. Two-tailed Student's *t* test was used for statistical analysis to compare the 6 h and 48 h timepoints. **C** Expression of select genes that have been annotated as a "Cytokine" or other pro-inflammatory genes by Ingenuity Pathway analysis (IPA). Plotted are log transformed TPM values that have been normalized across all 12 samples. Yellow indicates high gene expression; blue indicates low expression. Each columns represents an individual sample. **D** HSAEC1-KT cells were infected with *R. delemar* spores for 16 h, after which the levels of IL1α, IL1β, IL8, and GM-CSF protein levels were determined in the culture supernatants by ELISA. Values represent mean ± SEM of 2 experiments, each performed in biological triplicate. *n* = 6. Two-tailed Student's *t* test was used for statistical analysis.

We have previously demonstrated that URA of infection-induced differential gene expression can also be used to identify signaling pathways that facilitate the invasion of fungi onto epithelial cells[21,25,29]. This approach was validated by our identification of two pathways that are known to be involved in the interaction between epithelial cells and Mucorales: epidermal growth factor receptor (EGFR) and platelet-derived growth factor BB (PDGF BB) (Fig. 2A)[21,25]. Our analysis also predicted the activation of signaling pathways that have not been tied directly to mucormycosis including neural precursor-cell-expressed developmentally down-regulated protein 9 (NEDD9), G-protein coupled estrogen receptor 1 (GPER1), and hypoxia-inducible factor-1α (HIF1α)[29–33] (Fig. 2A). HIF1α signaling is of particular interest to us given its established role in angiogenesis[34–36] and the angio-invasive nature of mucormycosis. Invasion of blood vessels in the pathogenesis of

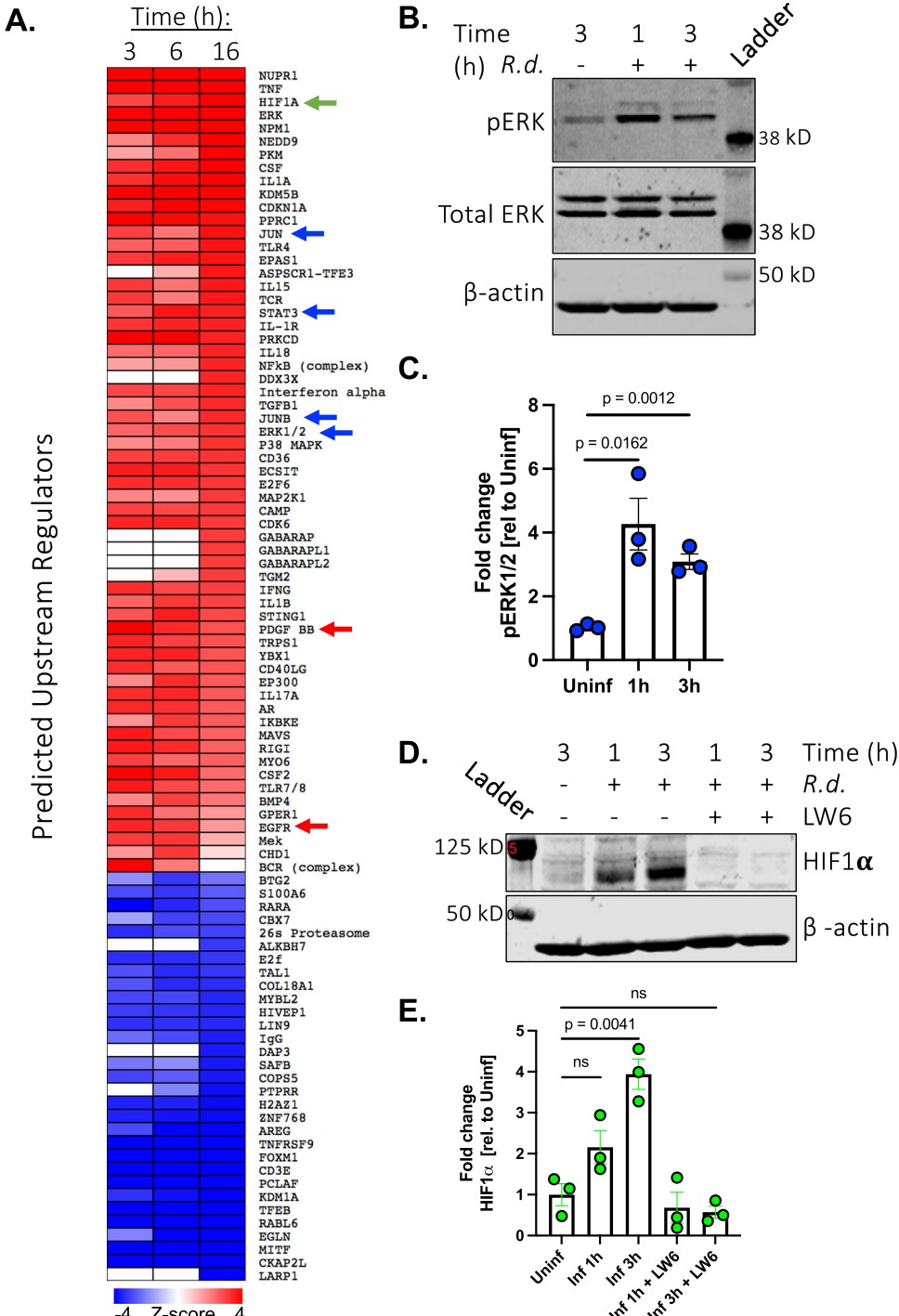

**Fig. 2 | Upstream regulator analysis and validation. A** Each regulator was predicted by Ingenuity Pathway Analysis (IPA) to be activated (red, Z-score >2) or repressed (blue, Z-score <2) during infection of HSAEC1-KT cells with *R. delemar* strain 99-880. Each of the depicted pathways have achieved a Z-score ≥ |3.0| in at least one of the timepoints. The complete analysis is displayed in Supplementary Data 5. White indicates no predicted activation or repression. Red arrows indicate pathways that were previously tied to *R. delemar* infection. The green arrow indicates a regulator of biological interest that was selected for functional follow-up experiments. Blue arrows indicate signaling pathways that are known to govern innate immune responses. **B** Representative immunoblot examining

phosphorylation of ERK1/2 in response to infection of HSAEC1-KT cells with *R. delemar*. **C** Densitometric analysis of the immunoblot in (**B**). Data represents the mean ± SEM of three independent experiments performed in singlicate. Two-tailed Student's *t* test was used for statistical analysis. **D** Immunoblot representing HIF1α accumulation from whole cell lysates collected 1 or 3 h post-infection of HSAEC1-KT cells with *R. delemar* in the presence or absence of HIF1α inhibitor, LW6. **E** Densitometric analysis of the immunoblot in (**D**). Data represents the mean ± SEM of three independent experiments performed in singlicate. Two-tailed Student's *t* test was used for statistical analysis. ns, not significant.

mucormycosis leads to tissue hypoxia, which would induce local production of HIF1α. Therefore, we chose to investigate the role of HIF1α during Mucorales infection.

### HIF1α is activated during *R. delemar* infection

The HIF1α protein is a transcription factor that facilitates the expression of hypoxia-responsive genes involved in angiogenesis, glucose metabolism, erythropoiesis, and apoptosis[37–39]. We observed a significant overlap between genes that are differentially expressed during *R. delemar* infection and the known transcriptional targets of HIF1α (*P* values of overlap, $4.51 \times 10^{-11}$, $5.2 \times 10^{-11}$, $3.85 \times 10^{-10}$ for 3, 6 and 16 h, respectively). Specifically, in vitro infection induced a change in the expression of 70 HIF1 target genes (66 genes whose expression is promoted by HIF1α and 4 whose expression is repressed by HIF1α), providing evidence for the activation of the HIF1α protein in response to *R. delemar* infection (Supplementary Fig. 1A).

HIF1α activation typically manifests as an increase in protein accumulation resulting from (i) an increase in mRNA transcription; (ii) an increase in protein translation or; (iii) an inhibition of ubiquitin-mediated protein degradation[40].

We used immunoblotting to determine if *R. delemar* infection leads to increased accumulation of HIF1α protein. As predicted by our RNA-seq analysis, we observed an increase in HIF1α protein abundance following in vitro infection of HSAEC1-KT cells with *R. delemar* (Fig. 2D, E). The expression of HIF1α transcript did not change over the first 3 h of the infection and decreased by 6 h post-infection indicating that the increase in protein accumulation was not a simple result of increased HIF1α mRNA production (Supplementary Fig. 1B). HIF1α accumulation can be pharmacologically inhibited by the addition of LW6, an aryloxyacetylamino-benzoic acid derivative that induces the expression of the von Hippel-Lindau (VHL) protein, which interacts with prolyl-hydroxylated HIF-1alpha to promote proteasomal degradation of HIF1α[41]. As expected, treatment of AECs with LW6 (20 μM) prevented infection-induced accumulation of HIF1α (Fig. 2D, E). Pretreatment of HSAEC1-KT cells with 25 μM gefitinib (a potent inhibitor or EGFR signaling), did not alter infection-induced accumulation of HIF1α (Supplementary Fig. 1C). Increased accumulation of HIF1α protein also occurred following infection of HSAEC1-KT cells with *Cunninghamella bertholletiae* (Fig. 3A, B) and *Rhizopus oryzae* (Supplementary Fig. 1D, E).

We next tested if *R. delemar* needs to be in direct contact with HSAEC1-KT cells to induce HIF1α accumulation. To this end, we measured HIF1α accumulation following infections with each cell type on opposite sides of a Transwell membrane which would allow for diffusion of small molecules but prevent contact with *R. delemar* (Fig. 3C). The separation of contact between the HSAEC1-KT cells and *R. delemar* by the Transwell membrane significantly reduced HIF1α accumulation (Fig. 3D, E), indicating that direct contact is necessary for the activation of the HIF1α pathway by *R. delemar*.

In a complementary approach, we performed indirect immuno-fluorescence microscopy of HSAEC1-KT cells to examine the protein levels and localization of HIF1α following in vitro infection with Mucorales. While HIF1α staining was apparent in both the cytoplasm and nuclei of uninfected AECs (Fig. 3F), nuclear HIF1α-derived fluorescent signal was significantly more intense in AECs that have been infected with *R. delemar* or *C. bertholletiae* for 3 h (Fig. 3F). Similar results were obtained following infection with isolates of 3 other pathogenic species of Mucorales (*R. oryzae*, *Mucor circinelloides*, and *Lichtheimia corymbifera*; Supplementary Fig. 2). These results are consistent with a model in which HIF1α protein levels increase and translocate to the nucleus in response to Mucorales infection.

Our previous analysis of the transcriptome response of A549 cells to infection with *R. delemar*, *R. oryzae* or *M. circinelloides* also predicted the activation of HIF1α signaling[25]. Taken together, these results support the idea that the HIF1α pathway is activated during infection of

epithelial cells by Mucorales and this activation is not a strain- or species-specific phenomenon.

### HIF1α facilitates the uptake of *R. delemar* by airway epithelial cells

We next performed a series of experiments to determine if the HIF1α pathway facilitates the interaction between *R. delemar* and small airway epithelial cells. We first studied the effects of LW6, an inhibitor of HIF1α accumulation (Fig. 2D, E), on endocytosis of *R. delemar* germlings. Treatment of HSAEC1-KT cells with LW6 (20 μM) significantly reduced the ability of *R. delemar* to invade HSAEC1-KT cells (Fig. 4A). We then used an siRNA knockdown approach to confirm the results of the small molecule inhibitor studies. Treatment of HSAEC1-KT cells with HIF1α-directed siRNA resulted in an almost complete block of HIF1α accumulation during *R. delemar* infection (Fig. 4B) and a >70% reduction in the ability of *R. delemar* germlings to invade the host cells (Fig. 4C). These results suggest that HIF1α facilitates the invasion of Mucorales into small airway epithelial cells.

### HIF1α facilitates *R. delemar*-induced innate immune responses in cultured small airway epithelial cells

Fan et al. demonstrated that HIF1α expressed in the intestinal epithelium of mice is required for the host defense against *Candida albicans* colonization and invasive disease[31]. To determine if HIF1α plays a role in the innate immune transcriptional responses elicited by *R. delemar* in small airway epithelial cells, we performed RNA-seq analysis following 3 h of in vitro HSAEC1-KT infection in the presence or absence of LW6. Our analysis also included time-matched uninfected samples that serve as negative controls. Each of the 3 experimental conditions were examined in triplicate. From each of the 9 sequencing libraries, we obtained an average of $281.7 \pm 20.9$ million reads that mapped to the human reference genome (Supplementary Data 1). We identified 16 pro-inflammatory genes whose expression changed significantly in the presence of LW6 (FDR < 0.01; $\log_2$ fold-change $\geq |1.0|$) including 12 genes with decreased expression and 4 genes with increased expression in the presence of LW6 (Fig. 4D, Supplementary Data 7). We chose 4 of these LW6-sensitive genes (LIF, EDN1, IL36G and CXCL8) for validation by quantitative RT-PCR on an independent group of infections (Fig. 4E). Indeed, our qRT-PCR results corroborated our RNA-seq analysis, thereby confirming that HIF1α is important for the ability of airway epithelial cells to mount a transcriptional innate immune response to *R. delemar* infection. Our observation that inhibition of HIF1α results in increased expression for some genes suggests a more complicated role in the host-response.

### The HIF1α pathway is activated during in vivo infection

We performed indirect immunofluorescence microscopy of lung tissue to examine the protein levels of HIF1α following 4 days of in vivo infection with *R. delemar* in our mouse model of mucormycosis. The fluorescent signal was significantly higher in the *R. delemar*-infected mice compared to the uninfected control mice (Fig. 5). Furthermore, HIF1α appeared to localize to the nuclei of the infected lung tissue as indicated by the overlap in HIF1α- and DAPI-derived fluorescence. These results support the idea that the HIF1α pathway is activated during in vivo infection by *R. delemar*.

### Inhibition of HIF1α increases survival of mice with mucormycosis

We next sought to determine if HIF1α facilitates disease establishment and/or progression in an in vivo murine model of mucormycosis. Mice harboring deletions in HIF1α suffer from severe cardiac and vascular malformations that result in death by embryonic day 10[42], thus preventing our ability to use a traditional mouse gene deletion experiment. Therefore, we infected neutropenic mice intratracheally with *R. delemar* and treated them with 15 mg/kg of LW6 or vehicle alone

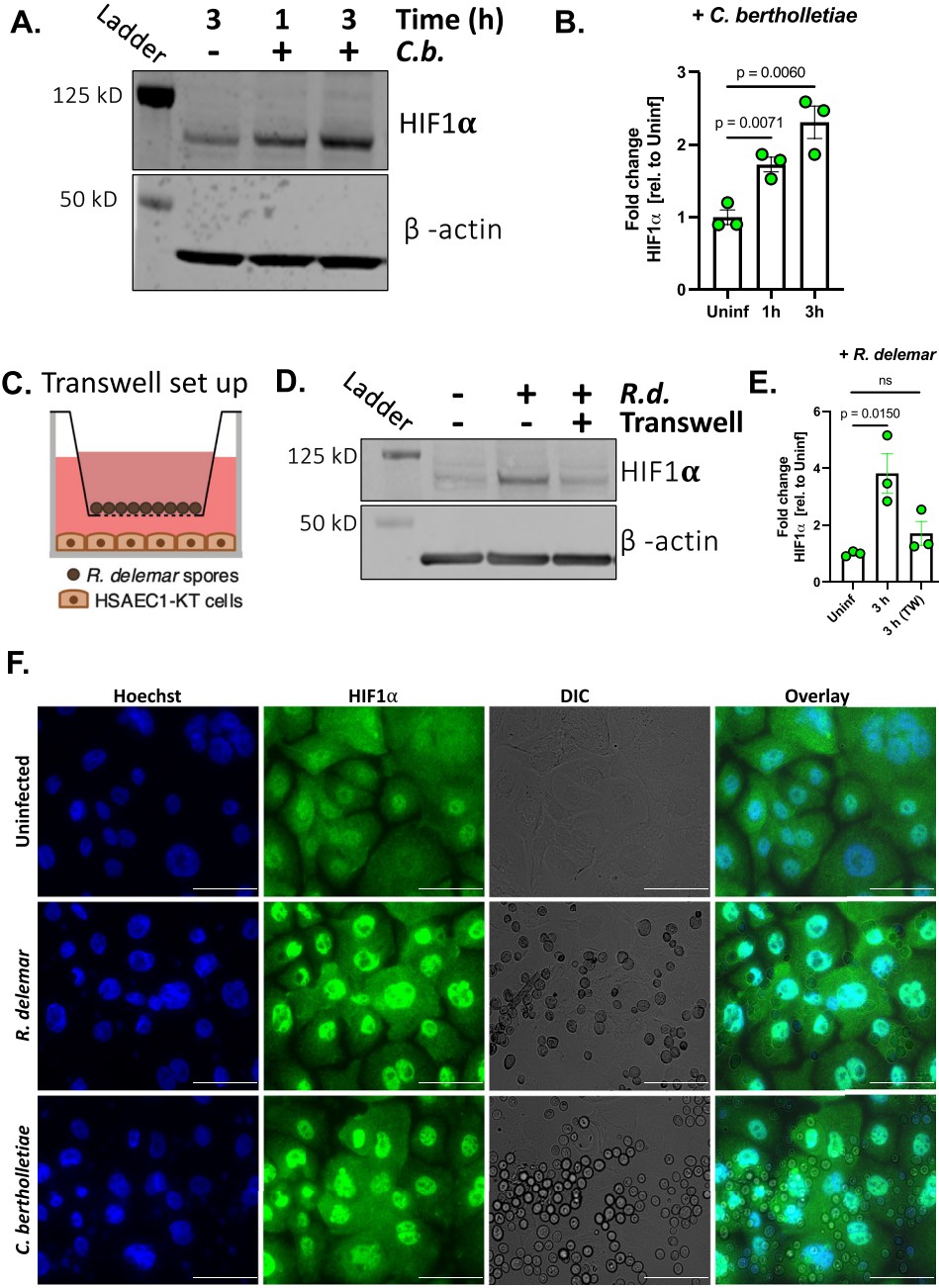

**Fig. 3 | *Mucorales* infection activates the HIF1α pathway in human airway epithelial cells. A** Immunoblot representing HIF1α accumulation from whole cell lysates collected 1 or 3 h post-infection of HSAEC1-KT cells with *C. bertholletiae*. **B** Densitometric analysis of the immunoblot in (**A**). Data represents the mean ± SEM of three independent experiments performed in singlicate. Two-tailed Student's *t* test was used for statistical analysis. **C** Schematic of transwell experiment. **D** Immunoblot representing HIF1α accumulation from whole cell lysates collected 3 h post-infection of HSAEC1-KT cells with *R. delemar* under standard conditions or

in a transwell. **E** Densitometric analysis of the immunoblot in panel D. Data represents the mean ± SEM of three independent experiments performed in singlicate. Two-tailed Student *t* test was used for statistical analysis. ns, not significant. **F** HSAEC1-KT cells were infected with *R. delemar* for 3 h and localization of HIF1α was assessed by indirect immunofluorescence with an anti-HIF1α antibody (green). Hoechst stain was used to visualize nuclei (blue). DIC, differential inference contrast. Scale bars = 100 μm. Microscopy experiments were performed three times independently with similar results.

(placebo) for 4 consecutive days starting 24 h post-infection. Placebo-treated mice had a median survival time of 7 days and 100% mortality by day 13 post-infection (Fig. 6A). In contrast, mice treated with LW6 had significantly lower HIF1α proteins levels in the lung tissue (Fig. 5) and a median survival time of 12 days, and 30% of the mice remained alive by day 21 (Fig. 6A).

In a complementary experiment, we measured the effects of LW6 on fungal burden in the lungs of infected mice. The fungal burdens of mice treated with LW6 were approximately 1 log higher than infected

mice that were treated with placebo (Fig. 6B). Taken together, these results indicate that inhibition of HIF1α function increases mouse survival without reducing fungal burden in the lungs.

## Discussion

In this work, we investigated the in vitro interaction between Mucorales and normal small human airway epithelial cell to understand the molecular processes that facilitate fungal invasion and innate immune responses by airway epithelial cells.

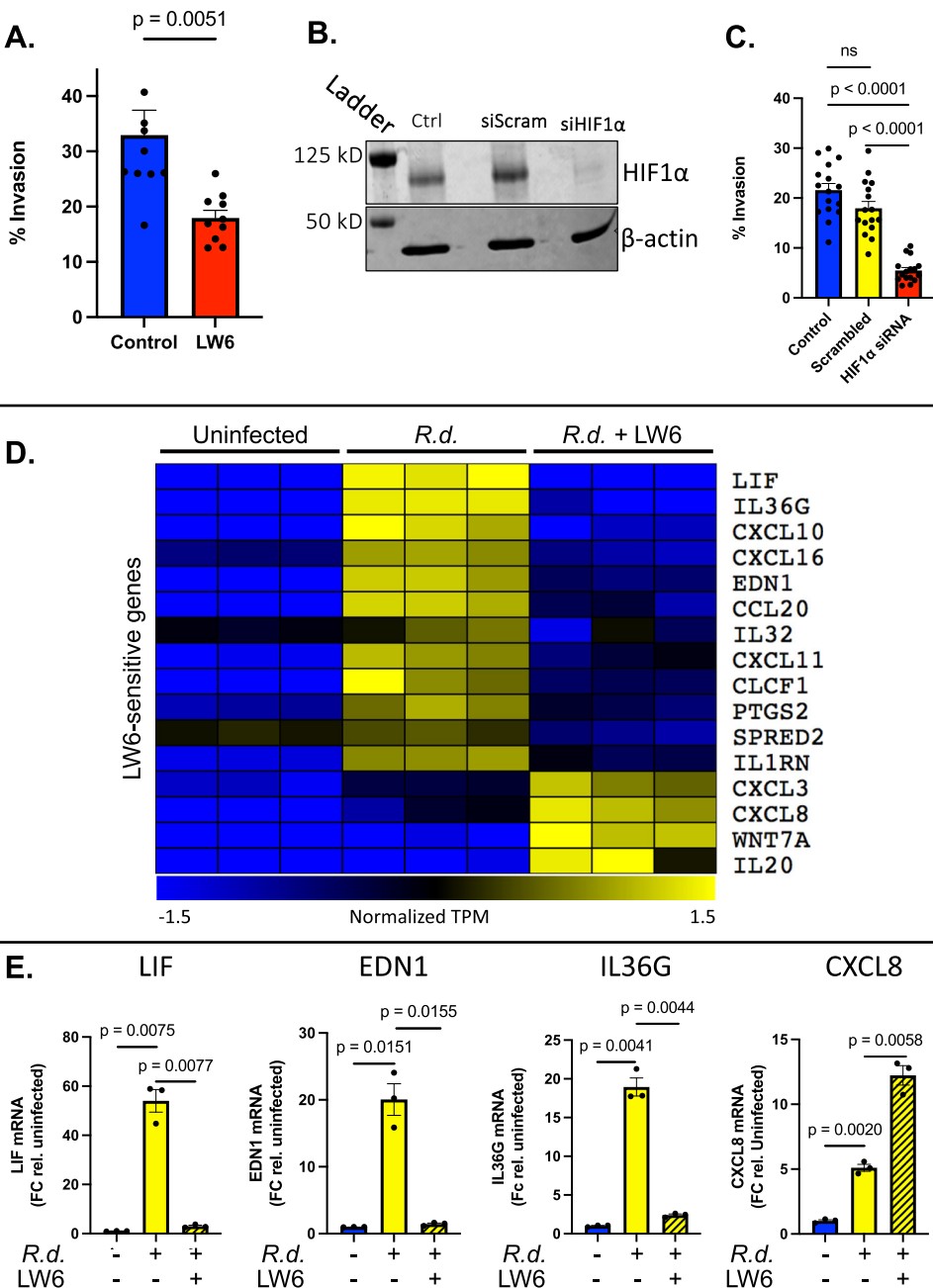

**Fig. 4 | Inhibition of HIF1α signaling in airway epithelial cells. A** Invasion of human small airway epithelial cells 3 h post-infection with *R. delemar* in the presence or absence of LW6 demonstrated by fluorescent microscopy. Values represent mean ± SEM of two independent experiments performed with 5 biological replicates each time. *n* = 10. Two-tailed Student's *t* test was used for statistical analysis. **B** Immunoblot representing depletion of HIF1α by siRNA knockdown. HSAEC1-KT cells were exposed to siRNAs for 48 h prior to infection, which was allowed to progress for 3 h. Experiment was performed two times independently with similar results. **C** Invasion of human small airway epithelial cells lacking HIF1α 3hrs post-infection with *R. delemar* demonstrated by fluorescent microscopy. Values represent mean ± SEM of two independent experiments performed with 8 biological replicates each time. *n* = 16. Two-tailed Student t-test was used for statistical analysis. NS, not significant. **D** Expression of select LW6-sensitive genes following 3 h of HSAEC1-KT cells infection with *R. delemar* in the presence or absence of HIF1α inhibitor, LW6. Plotted are log transformed TPM values that have been normalized across all 9 samples. Yellow indicates high gene expression; blue indicates low expression. Each column represents an individual sample. **E** mRNA levels for select genes determined by qRT-PCR performed on total RNA isolates. Data represents the mean ± SEM of three independent experiments performed in singlicate and independent of those depicted in panel (**D**). *R.d.*, *R. delemar*. Two-tailed Student t-test was used for statistical analysis.

The most salient findings of our study are (1) *R. delemar* infection induces HIF1α activation in airway epithelial cells; (2) HIF1α activation promotes fungal invasion into host cells; (3) HIF1α activation promotes a transcriptional innate immune response during *R. delemar* infection; and (4) inhibition of HIF1α increases survival in an in vivo murine model of mucormycosis.

The activation of HIF1α in airway epithelial cells and its role in fungal invasion into host cells has not been previously described. In the context of other fungal infections, HIF1α has been shown to be important for the immune response and fungal clearance by functioning in myeloid derived cells[43,44], macrophages[45], intestinal epithelial cells[31] and endothelial cells, but not previously in airway epithelial cells.

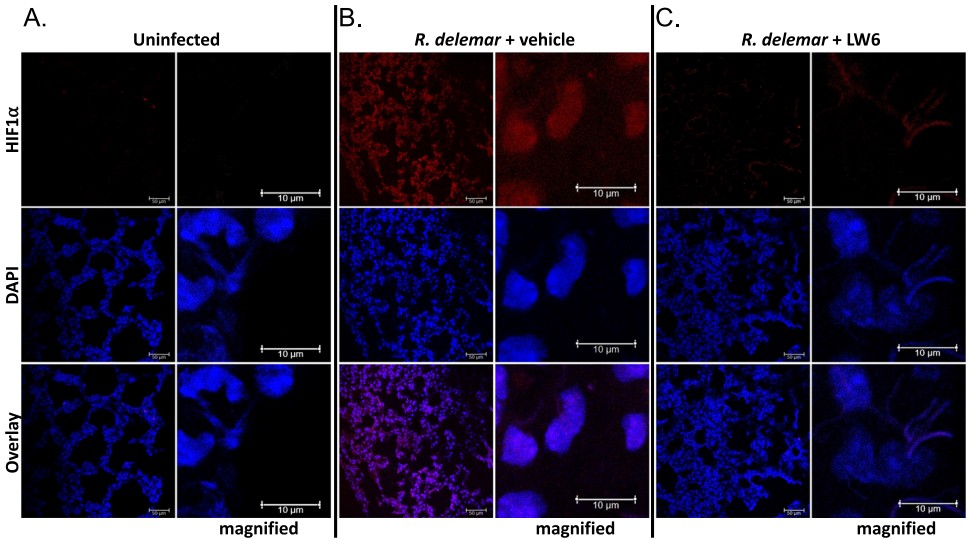

**Fig. 5 | *R. delemar* infection induces HIF1α accumulation in infected lungs.** Indirect immunofluorescence staining was performed on lung sections from uninfected mice (**A**), mice infected with *R. delemar* (**B**), and mice infected with *R. delemar* and treated with LW6 (**C**). Staining was performed with antibodies raised against HIF1α (red). The nuclei were stained with DAPI (blue). Scale bars = 50 μm; 10 μm in magnified images). Magnification = 8X. Experiment was performed two times independently with similar results.

Furthermore, we provide evidence that HIF1α plays two possibly opposing roles in the context of Mucorales infection in airway epithelial cells. One role in promoting fungal invasion into the host cells, a process that is strongly tied to Mucorales virulence[26,46,47], while the other role is to promote an innate immune response to clear the infection.

The incomplete block of invasion that we observed (Fig. 4A, C) indicates that the HIF1α pathway is not the only pathway to facilitate invasion into AECs. Indeed, we have previously reported that EGFR and Integrin α3β1 also govern Mucorales invasion into AECs[21,26]. Our observation that gefitinib treatment did not alter the infection-induced accumulation of HIF1α (Supplementary Fig. 1C) suggests that HIF1α activation occurs independently of EGFR signaling, however, further experiments are required to understand the interplay, if any, between HIF1α, EGFR and Integrin α3β1.

When cells are grown under conditions where oxygen is abundant (normoxia), HIF1α encoding mRNA is efficiently translated into a protein which is then rapidly hydroxylated by prolyl hydroxylases (PHD). This hydroxylation immediately targets HIF1α for ubiquitin dependent degradation by the von-Hippel-Lindau (vHL) protein. In the absence of oxygen, the hydroxylation, and subsequent degradation, is inhibited resulting in an increased accumulation of HIF1α protein levels in the cell. Once stabilized, HIF1α translocates to the nucleus where it forms a heterodimer with the HIF1ß protein. This heterodimer binds to hypoxia response elements (HREs) located in the promoters of hypoxia response genes to regulate their expression[37–39].

Following in vitro infection of our small airway epithelial cell lines, we observe a robust increase in HIF1α protein accumulation with no increase in HIF1α mRNA abundance implying that *R. delemar* infection might alter the stability or degradation of the protein. At this point, we do not know the molecular mechanism by which HIF1α signaling is being activated during *Rhizopus* infection. Further experiments are required to determine if the increased accumulation of HIF1α is a hypoxia-dependent process or a hypoxia-independent process as has been previously described[48–52], although the results of our Transwell experiments (Fig. 3D–F) suggest that the process is hypoxia-independent. Further study is also needed to characterize the nature of the innate immune response and whether it is protective or whether it may mediate host tissue injury. The impact of HIF1α on regulation of antimicrobial peptides by HSAEC1-KT airway cells against invasion by Mucorales also warrants further investigation[31].

In this study, our results show that LW6 inhibited *R. delemar* activation of HIF1α in HSAEC1-KT airway cells which resulted in inhibition of Mucorales-mediated invasion of the host cells, a process considered critical for the pathogenesis of invasive mucormycosis. Indeed, mice treated with LW6 had prolonged and overall survival vs placebo-treated mice. We acknowledge that a major limitation of our study is that the results of our in vivo experiments do not allow us to conclude that epithelial cell HIF1α is important during Mucorales infection since the LW6 is likely to inhibit HIF1α expressed in other cell types. It is possible that by prevention of host cell invasion into airway epithelial cells, LW6 reduces *R. delemar* dissemination and prevents wide-spread infection which results in death[53]. It is also possible that mice infected with *R. delemar* die from an exuberate immune response and dampening this immune response by LW6 might help the mice to survive the infection. The lack of reduction of lung fungal burden in mice treated with LW6 is somewhat expected given the role of HIF1α in stimulating innate immunity, a response critical in clearing the infection[44]. This lack of reduction in lung fungal burden coupled with enhance survival in LW6-treated over placebo mice also emphasizes the role of host cell invasion in determining the outcome of the disease in this animal model. The mechanism by which LW6 protects mice from mucormycosis and the possibility of its use as an adjunctive therapy is currently under active investigation.

Case reports indicate some benefit in treating rhino-orbital/cerebral mucormycosis patients with hyperbaric oxygen[54–56]. It is theorized that hyperbaric oxygen would provide oxygenation of tissues thereby reducing acidosis that aids in pathogenesis of mucormycosis, and oxygen in sufficient concentration is fungicidal[54]. Our studies may indicate that patients with pulmonary mucormycosis might benefit from such treatment by reducing hypoxia and activation of the HIF1α pathway.

In summary, our data show that Mucorales can invade small human airway epithelial cells in vitro through the activation of the HIF1α transcription factor resulting in a hypoxia-induced activation of host immune response. Inhibition of Mucorales-induced HIFα activation protects HSAEC1-KT airway cells and mice from mucormycosis.

## Methods
### Ethics Statement
All research was performed in compliance with all ethical regulations under IBC-00003284 approved by the Institutional

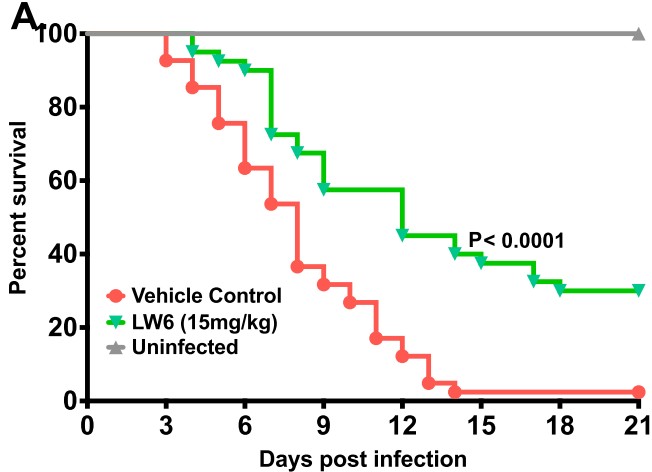

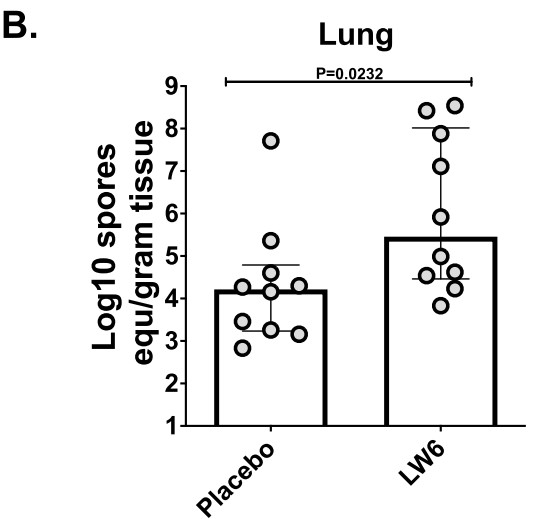

**Fig. 6 | LW6 protects mice from pulmonary mucormycosis. (A** Survival of neutropenic mice (40 per group from three experiments) infected intratracheally with *R. delemar* (average inoculum of 1.1×10^4 spores per mouse) and treated with vehicle control (placebo) or 15 mg/kg LW6 24 h post infection for 4 consecutive days. (**B**) Inhibition of the HIF1α by LW6 did not affect the fungal burden in the lungs of mice harvested on Day +4 post infection. Data from one experiment (10 mice per group) and presented as median + interquartile range. Statistical analysis was performed by using Mann-Whitney non-parametric (two-tailed) test comparing treated groups *vs*. placebo group.

Biosafety Committee at the University of Maryland, Baltimore. All animal studies were approved by the Institutional Animal Care and Use Committee (IACUC) of the Los Angeles Biomedical Research Institute at Harbor-UCLA Medical Center according to the NIH guidelines for animal housing and care (approval reference no. 21125).

## Fungal strains and culture conditions

*Rhizopus delemar* strain 99-880 (a clinical isolate obtained from a patient with rhino-orbital mucormycosis), *Rhizopus oryzae* strain 99-892, *Lichtheimia corymbifera* strain 008-049, *Mucor circinelloides* strain NRRL3631, and *Cunninghamella bertholletiae* strain 175 were grown on peptone-dextrose agar (PDA) plates for 3 to 5 days at 37 °C. Spores were collected in endotoxin-free Dulbecco's phosphate-buffered saline (DPBS), washed with endotoxin-free DPBS, and counted with a hemocytometer to prepare the final inocula.

## Cell culture and transient transfection

HSAEC1-KT cells (CRL-4050, ATCC) were maintained in flasks with SABM Basal Medium (Lonza CC3119) supplemented with SAGM SingleQuots (Lonza CC4124) at 37 °C, 5% $CO_2$. The cells were fed every 2–3 days and passaged once a week. For transfection, cells were plated at a seeding density of $0.25 \times 10^6$ cells/ml to reach a confluency of 80–90% on the day of transfection. Xfect (Cat# 631450, Takara Clontech) was used to transfect cells with 200 pmol siRNAs or with control siRNA. The growth medium was replaced 4-hr post-transfection. Cells were harvested 48 h post-transfection for downstream experimental analysis.

## *R. delemar* induced damage to human small airway epithelial cells (HSAEC)

A chromium ($^{51}$Cr)-release assay was used to quantify the host cell damage[57]. Briefly, human small airway epithelial cells (HSAEC) grown in 96-well tissue culture plates were incubated with 1 µCi per well of $Na_2{}^{51}CrO_4$ (ICN) in SAGM medium for 16 h. On the day of the experiment, the wells were washed twice with warm HBSS. HSAE Cells were infected with $5 \times 10^4$ spores suspended in 200 µl SAGM medium. Spontaneous $^{51}$Cr release was determined by incubating HSAEC in SAGM medium without *R. delemar* for each time point. After 6, 12, 24, 36, 48 h of incubation of germlings with HSAEC, 50% of the medium was aspirated from each well and transferred to glass tubes. The other 50% of the medium with the cells was transferred to another glass tube. The amount of $^{51}$Cr in the tubes was determined by gamma counting. The total amount of $^{51}$Cr incorporated by HSAEC in each well equaled the sum of radioactive counts per minute of the aspirated medium plus the radioactive counts of the corresponding cells. After the data were corrected for variations in the amount of tracer incorporated in each well, the percentage of specific epithelial cell release of $^{51}$Cr was calculated by the following formula: $\left[\frac{(\text{Experimental release} - \text{Spontaneous release})}{(\text{total incorporation} - \text{spontaneous release})}\right]$. Each experimental condition was tested at least 6 replicates, and the experiment was repeated two times.

## RNA extraction, RNA-seq and gene expression analysis

RNA-seq experiment # 1 examined *R. delemar*-induced gene expression at 3, 6, and 16 h post-infection (Supplementary Data 1). RNA-seq experiment # 2 examined LW6-sensitive gene expression following 3 h of infection with *R. delemar* (Supplementary Data 1). Total RNA was isolated from HSAEC1-KT cells post-infection using the PureLink RNA isolation kit (Invitrogen #12183020). RNA-seq libraries (strand-specific, paired end) were generated from total RNA by using a TruSeq RNA sample prep kit (Illumina). One hundred nucleotides of sequence were determined from both ends of each cDNA fragment using the Novaseq platform (Illumina). Sequencing reads were aligned to the human reference genome (GRCh38 release 101) using HISAT2[58], and alignment files were used to generate read counts for each gene. Statistical analysis of differential gene expression was performed using the DEseq package from Bioconductor[59]. A gene was considered differentially expressed if the absolute log fold change was greater than or equal to 1 and the FDR value for differential expression was below 0.01. The RNA-seq analysis was performed in biological triplicate. The normalized reads counts for each sample of RNA-seq experiment #1 are listed in Supplementary Data 6. The normalized reads counts for each sample of RNA-seq experiment #2 are listed in Supplementary Data 8.

For qRT-PCR follow-up experiments, 500 ng total RNA was reverse transcribed using Maxima H Minus cDNA Synthesis Master Mix (M1681, Thermo Scientific). Quantitative PCR (qPCR) was performed using KAPA SYBR FAST qPCR Master Mix Kit (KK4601, Kapa Biosystems) and 10 µM each of forward and reverse transcript-specific primers, using a CFX 384 real time system (Bio-Rad). The program consisted of one cycle of 95 °C for 3 min, 40 cycles of 95 °C for 3 s, 60 °C for 40 s (annealing and polymerization) followed by melting

curve analysis. Data were analyzed using CFX Maestro software (Bio-Rad). Relative mRNA levels for genes were normalized to the geometric mean of the endogenous controls GAPDH, and RPL13A using the 2^-ΔΔCt method[60].

To identify signal transduction pathways that are activated or repressed during our in vitro infections, we used the upstream regulator analytic (URA) of IPA (Ingenuity Systems; http://www.ingenuity.com). This analysis determines the overlap between lists of differentially expressed genes and an extensively curated database of regulator-target gene relationships. The direction of gene expression change is then considered to make predictions about pathway activation (z-score >2) or repression (z-score < −2).

### Cytokine analysis

Cytokines were measured by two-antibody ELISA using biotin-streptavidin-peroxidase detection. Polystyrene plates (Maxisorb; Nunc) were coated with capture antibody in PBS overnight at 25 °C. The plates were washed 4 times with 50 mM Tris, 0.2% Tween-20, pH 7.0-7.5 and then blocked for 90 min at 25 °C with assay buffer (PBS containing 4% BSA (Sigma)). Then 50 μl of sample or standard prepared in assay buffer was incubated at 37 °C for 2 h. The plates were washed 4 times and 100 μl of biotinylated detecting antibody, in assay buffer, was added and incubated for 1 h at 25 °C. After washing the plate 4 times, streptavidin-peroxidase polymer in casein buffer (RDI) was added and incubated at 25 °C for 30 min. The plate was washed 4 times and 100 μl of commercially prepared substrate (TMB; Dako) was added and incubated at 25 °C for approximately 10–30 min. The reaction was stopped with 100 μl 2 N HCl and the A450 (minus A650) was read on a microplate reader (Molecular Dynamics). A curve was fit to the standards using a computer program (SoftPro; Molecular Dynamics) and cytokine concentration in each sample was calculated from the standard curve equation.

### Endocytosis assay

The percentage of endocytosed fungal spores into HSAEC1-KT cells were determined by endocytosis assay as previously described by Watkins et al.[21]. Briefly, cells were plated onto 25 mm Poly-L-Lysine Coated German Glass Cover Slips (Cat#72292-20, EMS) and placed at $CO_2$ incubator at 37 °C for 5–7 days till it reached 80-90% confluency. Following 3 hrs of infection with swollen fungal spores, the cells were washed with endotoxin free DPBS and fixed with 3% paraformaldehyde (Cat# 50-980-494, Fisher Scientific). After fixation the infected cells were washed once with endotoxin free DPBS and stained with 1% Uvitex solution (NC9859220, Polysciences), which specifically binds to the chitin of the fungal cell wall. The coverslips were then dipped in a beaker containing DPBS (14190144, Invitrogen) for 5–8 times followed by 5-8 washes in beaker containing molecular biology grade water. The coverslips were then mounted on a clean glass slide with 20ul Prolong Gold antifade reagent (P36930, Thermo Fisher Scientific) and sealed with nail polish. These slides were imaged in Keyence BZ-X700 fluorescent microscope. The total number of cell-associated spores was determined by phase-contrast microscopy. The same field was examined by epifluorescence microscopy, and the number of uninternalized spores were determined. The number of endocytosed spores was calculated by subtracting the number of fluorescent spores from the total number of visible spores. At least 1000 spores were counted in 8 different fields on each slide. Fungal overgrowth precluded our ability to use this assay to assess fungal invasion at later timepoints (Supplementary Fig. 3).

### Western blot

Following infection, cells were rinsed with cold endotoxin free DPBS containing 1 mM $Na_3VO_4$. Cell lysates were prepared by incubating the cells for 5 min on ice in 500ul ice cold cell lysis buffer containing 1X Cell Lysis Buffer (Cell Signaling #9803),1X protease Inhibitor Cocktail Set V

(EMD Millipore Corp, Cat# 539137), phosphatase Inhibitor Cocktail 2 (Sigma P0044) and Cocktail 3 (Sigma P5726). The cells were then scrapped from the plate and sonicated to generate the cell lysates. Following sonication, cell lysates were centrifuged for 10 min at 4 °C. The supernatant was collected and stored at −80⁰C. The cell lysates were then subjected to electrophoresis using XCell SureLock Mini-Cell II (Life technologies) and blotted on a polyvinylidene difluoride membrane using Invitrogen XCell II Blot Module (Life technologies) as per manufacturer's recommendations. The membranes were blocked with TBS blocking buffer (Licor) for 1-hr prior to overnight incubation with primary antibodies. The anti-HIF1α antibody (20960-1-AP) was purchased from Proteintech and used at a 1:1000 dilution. The anti-β-actin (8H10D10), the anti-ERK1/2 (137F5), and anti-phospho-ERK1/2 (D13.14.4E) antibodies were purched from Cell Signaling Technologies and used at a 1:1000 dilution. On the following day, the membranes were washed four times with TBS-t (1X TBS containing 0.1% Tween 20) followed by 1 h incubation at room temperature with IRDye secondary donkey antibodies. The proteins were then visualized using Odyssey CLx Imaging system and quantified with Image Studio software. Where indicated, 22 mm transwells with 0.4 μm pore polyester membrane inserts (Costar #3450) were used to separate the fungus from the HSAEC1-KT cells.

### Immunoflourescence microscopy

For in vitro infections, HSAEC1-Kt cells were seeded onto poly-L-lysine coated circular glass coverslips (EMS) in 6-well tissue culture plates and incubated at 37 °C in 5% $CO_2$. The cells were infected with swollen fungal spores at MOI 1:5 (approximately $10^6$) in EGF-free SABM media (Lonza) for 3 or 6 h. After aspirating the media, cells were washed 3 times with PBS, fixed with 4% paraformaldehyde for 10 min and washed with PBS. Cells were permeabilized with 0.25% Triton X-100 (Sigma-Aldrich) in Tris buffered saline (TBS), 1% BSA for 10 min. Cells were blocked with 5% of BSA in TBS for 1 h. Cells were labeled with rabbit anti-HIF1α (Proteintech, Catalog #:20960-1-AP) followed by secondary AlexaFluor 488 labeled goat anti-rabbit IgG antibody (ThermoFisher). Nuclei were visualized by labeling with Hoechst 33342 (ThermoFisher). The slides were mounted using ProLong Gold Antifade mountant (ThermoFisher). Slides were imaged by confocal microscopy using EVOS MT700 imaging system (Invitrogen). Multiple images were obtained along the z-axis and stacked using EVOS MT7000 software (Invitrogen).

For the in vivo infections, mice were sacrificed at day 4 post infection and cold PBS was used to remove the blood from the lung tissues which were then treated with OCT embedding compound. The tissues were frozen on dry ice and stored at −80 °C until sectioning into 10 micron slices. The tissues were fixed with cold methanol (−20 °C) for 20 min. Slides were rinsed twice with 10 mM phosphate buffered saline (PBS) at a neutral pH followed by addition of blocking buffer (1% goat serum in PBS) and incubation in a humidified chamber at room temperature for 1 h. The blocking buffer was drained from the slides, Biotin-labeled mouse anti-HIF1α monoclonal antibody (Bio-Techne Corporation) was applied, and the slides were incubated overnight at 4 °C in a humidified chamber. The antibody was biotin-labeled using the EZ-Link™ Sulfo-NHS-LC-Biotinylation Kit (Thermo Scientific). The slides were rinsed twice with PBS for 5 min each. Alex Fluor 568 conjugated Streptavidin (Invitrogen, catalog: S11226) was added to the sections on the slides followed by incubation in a humidified chamber, in the dark, at room temperature for 30 min. The slides were rinsed twice again for 5 min each. Mounting media with DAPI (Abcam) was applied and a coverslip was placed over the sections. The slides were imaged by confocal microscopy.

### Murine models of mucormycosis

To test the effect of LW6 on mouse survival following infection, male ICR mice (20 to 25 g [from Envigo]) were immunosuppressed by

cyclophosphamide (200 mg/kg administered intraperitoneally [i.p.]) and cortisone acetate (500 mg/kg administered subcutaneously) given on days −2, −3, and 8 relative to infection. This treatment resulted in 16 days of pancytopenia. To control for bacterial infection, immuno-suppressed mice received 50 mg/liter enrofloxacin (Baytril; Bayer, Leverkusen, Germany) *ad libitum* on day -3 through day 0, after which the enrofloxacin was replaced with daily ceftazidime (5 mg/mouse) treatment administered subcutaneously through day 13 relative to infection. Mice were infected with 2.5 x 10$^5$ spores of *R. delemar* 99-880 in 25 μl PBS given intratracheally as previously described[21]. Treatment with LW6 (15 mg/kg dissolved in vehicle solution, containing 10% dimethylacetamide, 10% Cremophor EL and 80% of sodium carbonate buffer and administered p.o.) started 24 h post-infection and continued twice daily through day 4. Placebo-treated mice received vehicle solution only by p.o. Survival of mice served as the primary endpoint, with moribund mice humanely euthanized. To determine the effect of treatment on tissue fungal burden, mice were immuno-suppressed and infected as described above. LW6 treatment started 24 h post-infection and continued through day 4 at which point the mice were sacrificed, and lungs and brains, representing primary and secondary target organs, were collected and processed for tissue fungal burden by quantitative PCR (qPCR) using oligonucleotide amplification primers of the *R. arrhizus* 18S rRNA gene.

(GenBank accession no. AF113440) as previously described[61]. The specific sequences of these oligonucleotides are as follows: (i) sense amplification primer, 5′-GCGGATCGCATGGCC-3′; and (ii) antisense amplification primer, 5′-CCATGATAGGGCAGAAAATCG-3′. Values are expressed as log10 spore equivalents per gram of tissue.

## Statistics and reproducibility

No statistical tests were utilized to pre-determine sample size. As it is the minimum number of replicates required for inferential analysis, at least three biological replicates were utilized for all experiments. No data were excluded from analyses, the experiments were not rando-mized, and investigators were not blinded to allocation during experiments of outcomes assessment. Statistical analyses were per-formed using GraphPad Prism 10.2.0 for Windows (GraphPad Soft-ware, SanDiego, CA, USA). Specific tests used to determine statistical analyses are noted in each figure legend. *p* values are depicted, with a value of *p* < 0.05 considered significant.

## Reporting summary

Further information on research design is available in the Nature Portfolio Reporting Summary linked to this article.

## Data availability

All of the raw sequencing reads from this study are available at the NCBI Sequence Read Archive (SRA) under BioProject accession num-ber PRJNA986553. The specific sample accession numbers are listed in Supplementary Data 1. Source data are provided with this paper.

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

## Acknowledgements

This work was supported by NIH grants U19AI110820 and R01AI141360 (V.M.B., A.S.I.). T.J.W. was supported by the Henry Schueler Foundation. P.K. is a Doctoral Scholar of the Henry Schueler Foundation. A.S.I is supported by NIAID grant R01AI063503. RNA-sequence data was generated by Maryland Genomics at the Institute for Genome Sciences, University of Maryland School of Medicine.

## Author contributions

Conceptualization and planning: P.K., Y.G., N.H., T.J.W., A.S.I. and V.M.B.; Experimentation: P.K., Y.G., N.H., K.T.G. and A.A.; Data analysis: P.K., A.C.S., C.M., T.J.M., A.S.I. and V.M.B.; Manuscript and figures preparation: P.K., Y.G., N.H., A.A., T.J.M., A.S.I. and V.M.B.

## Competing interests

A.S.I. owns shares in Vitalex Biosciences, a start-up company that is developing immunotherapies and diagnostics for mucormycosis. T.J.W. has received grants for experimental and clinical antimicrobial pharmacology, therapeutics, and diagnostics to his institution from Amplyx, Astellas, Gilead, Lediant, Merck, Scynexis, Shionogi, T2 Biosystems, Viosera; and has served as a consultant to Abbott, Astellas, Karyopharm, Leadiant, Partner Therapeutics, Scynexis, Shionogi, Statera, and T2 Biosystems. The remaining authors declare no competing interests.
