## [Peer Review File · Nature Communications]

Multiple roles for Hypoxia inducible factor 1-alpha in airway epithelial cells during mucormycosisREVIEWER COMMENTS

Reviewer #1 (Remarks to the Author):

NCOMMS-23-27702 comments

Rhizopus delemar is an emerging opportunistic fungal pathogen often causing devastating mucormycosis in individuals with compromised immune status. Still, the mechanisms of the disease and host-pathogen interaction are not understood. An earlier study using an in vitro model of *R. delemar* and alveolar basal epithelia A549 cells suggested that the interaction between the two is vital for the disease whose process may involve the EGFR pathway of host cells. Here, Hasin and nine other colleagues re-examined the interaction using a different epithelial cell line, HSAEC1-KT, and found that *R. delemar* provokes a pro-inflammatory response. The authors found that, among several candidate host signaling pathways, the hypoxia-responsive transcription factor HIF1 α appeared to govern the *R. delemar*-host interaction. Further studies suggested that regulation by HIF1 α is independent of transcription and that an inhibitor of HIF1 α , LW6, may modulate mouse survival in a murine pulmonary infection model.

Understanding disease mechanisms and host responses to fungal pathogens is critical for properly managing or treating mucormycosis infections. Therefore, the topic of the work is important. However, the work appears to be looking at a general host response that lacks any specificity. Some data showing either non-impressive or hard to interpret results suggests inadequacy in experimental approaches. While this study adds to current knowledge on this group of understudied fungal pathogens, the narrow scope of work and incremental but often conflicting findings make it unsuited to the types of audience NCOMMS is expected to attract.

Examples of issues that are of concern include:

1) "Overall lower invasion and damage of HSAEC-1 KT cells than alveolar and nasal cells": the authors did not include any of these cell lines as controls in their studies, so the conclusion may not be accurate.

2) "No bead-beating steps in RNA extraction to reduce fungal inputs": the method may not be scientifically rigorous. RNA readings may still need to map against a fungal database(s) to eliminate those of the fungus origin, even though no database is specific to Mucorales.

3) "These results suggest that HIF1 α governs the invasion of Mucorales into small airway epithelial cell": without several host signaling pathways that are potentially involved in the *R. delemar*-HSAEC-1 KT interaction and the authors examined only one such pathway with marginally significant findings, this appears to be an overstatement.

4) There are variances in RNA-Seq data between Fig 2. A and Fig 5. A, particularly in between SPRED2, IL32, and IL20, which the authors failed to discuss.

5) LW6 in vivo: there is no difference in mean survival and essentially no difference in total survival ($p = 0.36$) between no treatment and 5 mg/kg treatment. Results from an increased dosage (increase of 2 days in mean survival and 40% compared to 30% at 5 mg/kg) still did not appear remarkable. Compounding the issue was that no change in fungal burden was detected. The uncertain results could only suggest that 1) the role of LW6 in governing fungal infection is marginal in addition to being indirect; 2) the experimental approach is inappropriate, which includes the authors' methods of estimating fungal burden with qPCR and expressing the burden as log₁₀ spore/tissue weight in an infection by a filamentous fungus.

Reviewer #2 (Remarks to the Author):

The manuscript "Multiple roles for Hypoxia inducible factor 1-alpha in airway 1 epithelial cells during mucormycosis" by Hasin et al. explores the in vitro transcriptomic response of a cell line derived from small airway cells. This analysis reveals a significant number of differentially expressed genes (DEGs), and subsequent examination of upstream regulators identifies potential candidates involved in regulating the response of epithelial cells to *Rhizopus delemar* invasion. The authors specifically focus on HIF-1 α and explore its role in epithelial cell invasion, damage, and in vivo virulence. The most notable finding is the identification of the link between HIF-1 α and mucormycosis, particularly in relation to the invasion of airway epithelial cells, immunity, and virulence. The information derived from this study may hold great importance in the field and has the potential to become a landmark publication. However, there are several significant concerns that need to be addressed to firmly establish the role of HIF-1 α in mucormycosis and justify its publication in a high-impact journal such as Nature Communications.

Major concerns

1. The specificity of the transcriptomic response of epithelial cells to *R. delemar* is uncertain due to the absence of a control group in which cell damage is induced through alternative means. It is recommended to compare the transcriptomes of cells damaged by different methods to distinguish between the response mediated by the fungus and the response resulting from general cell damage.
2. Verification of the *R. delemar*-mediated induction of innate immune response in cultured small airway epithelial cells should be performed at the protein level.
3. The selection of HIF-1 α as the primary candidate involved in the response to the fungus raises concerns as several other genes identified in Figure 2B could potentially govern the response of epithelial cells to *R. delemar*. It should be clearly stated that multiple genes may contribute to the observed cellular response.
4. The relevance of Figure 3A appears to be low, and it is recommended to move it to supplementary material since the results are expected based on the upstream regulatory analysis (URA) that selected HIF-1 α due to the enrichment of its targets in the transcriptomes of cells exposed to the fungus. Additionally, information regarding the datasets used in this analysis is missing.
5. It is suggested to expand the experiments analyzing the conservation of HIF-1 α activation to include more species, considering the close phylogenetic relationship between *R. delemar* and *R. oryzae*. Simple western experiments with cultured cells can be repeated using at least *Cunninghamella*, *Lichtheimia*, *Rhizomucor*, and *Mucor*. Quantification of the Western results with a non-saturated loading control is recommended, as the signal of β -actin is saturated, making it difficult to estimate changes in HIF-1 α , particularly when the differences in the signal for HIF-1 α protein are small.
6. An explanation regarding how *R. delemar* activates HIF-1 α is crucial to understanding the results presented in Figure 4B and 4C. These figures can be better interpreted if we consider that HIF-1 α activation occurs before spore endocytosis, explaining why the failure in activation reduces invasion. Furthermore, conducting experiments to investigate the effect of HIF-1 α upregulation on Mucorales invasion could support the hypothesis proposed by the authors. To gain further insights into the link between HIF-1 α activation and endocytosis, experiments with knockdown strains of *R. delemar* for *coH7* and $\alpha 3\beta 1$ -depleted epithelial cells should be conducted, as a reduction in endocytosis is expected.
7. Confirmation of the increase in HIF-1 α in the response of airway epithelial cells to Mucorales spore endocytosis or contact should be performed in vivo through, for example, immunohistochemical analysis.
8. A more comprehensive description of the results obtained from the RNA-seq analysis using the LW6 inhibitor is necessary to gain insights into the significance of the differential expression of 16

pro-inflammatory genes in response to LW6.

9. The virulence assays shows only a slight reduction in virulence with the highest dose of LW6. More conclusive experiments are required, such as using mouse with tissue-specific deletion of HIF-1 α (Saini Y, Greenwood KK, Merrill C, Kim KY, Patial S, Parameswaran N, Harkema JR, LaPres JJ. Acute cobalt-induced lung injury and the role of hypoxia-inducible factor 1 α in modulating inflammation. *Toxicol Sci.* 2010 Aug;116(2):673-81. doi: 10.1093/toxsci/kfq155) and HIF-1 α upregulation (Su H, Yi J, Tsui CK, Li C, Zhu J, Li L, Zhang Q, Zhu Y, Xu J, Zhu M, Han J. HIF1- α upregulation induces proinflammatory factors to boost host killing capacity after *Aspergillus fumigatus* exposure. *Future Microbiol.* 2023 Jan;18:27-41. doi: 10.2217/fmb-2022-0050)

Minor points

1. Consider adopting the nomenclature proposed by Dolatabadi et al, 2014 (Dolatabadi S, de Hoog GS, Meis JF, Walther G. Species boundaries and nomenclature of *Rhizopus arrhizus* (syn. *R. oryzae*). *Mycoses.* 2014 Dec;57 Suppl 3:108-27. doi: 10.1111/myc.12228).
2. In line 94, consider replacing "conidia" with "spores", which is a more general term.
3. In lines 131-133, to draw this conclusion, experiments should be conducted under the same conditions, in the same laboratory, and at the same time.
4. Supplementary Tables 2, 3, and 4 should include normalized reads for each sample.
5. The meaning of red circles is described twice in the legend of Figure 2.
6. The fungal strains used in this work should be specified.

Reviewer #3 (Remarks to the Author):

In this manuscript, Hasin and colleagues investigated the role of the transcription factor hypoxia-inducible factor 1 alpha (HIF-1 α) during mucormycosis. The authors propose that HIF-1 α plays a dual role during infection, one involving the control of host cell invasion by the fungus and the other the regulation of the immune response to infection. While this study represents an important starting point in the characterization of the role of HIF-1 α in the regulation of epithelial cell function in response to mucormycosis, it mostly describes HIF-1 α induction and the effects of its inhibition on host cell invasion by the fungus, with limited mechanistic data as to why HIF-1 α might have a dual role during infection, as suggested by the authors. I have several comments that may help improve the manuscript.

1. During the invasion experiments, data is shown until 6 h after infection. Does invasion still increase at later time points, since cell damage does rises up until 48h? Also, to what extent is damage due to the fungal invasion of epithelial cells versus the presence of extracellular fungi that eventually germinate? Is there any evident fungal growth? In this regard, the authors might consider including representative microscopy images of the infected cells at relevant time points.
2. Regarding the RNAseq experiments, the authors mention that analysis was performed on host transcripts since no lysis of the fungal cell wall was performed and therefore no fungal RNA was available. However, an important control should be provided by showing the expression of a fungal target (e.g., 18S) and determining the relative ratio of human to fungal RNA.
3. Given that invasion of epithelial cells is known to rely on the interaction of CoH7 in Mucorales spores with the integrin α 3 β 1 in host cells, the question arises as to whether this interaction is required for the activation of HIF-1 α . Along the same line, is EGFR signaling required for HIF-1 α induction? This is particularly relevant since, as the authors indicate, EGFR was identified as a predicted upstream regulator of HIF-1 α .
4. Another important aspect of epithelial cell function during infection and that was not addressed – even though the authors recognize its relevance in the discussion – is the production of antimicrobial peptides and the extent to which this is regulated by HIF-1 α activation in response to infection.

5. The immunoblotting of HIF-1a demonstrated major differences in the cellular amounts of the transcription factor. However, while this may be indicative, together with the expression of target genes for HIF-1a, it hardly confirms the activation of HIF-1a in response to infection. Immunoblotting should instead be performed in cytosolic and nuclear extracts in order to allow the quantification of nuclear translocation of HIF-1a and therefore its activation.
6. Along the same line, the analysis of both protein and mRNA expression should also include later time points. Given that mRNA starts to decrease at 6h, it would be important to know whether total protein also decreases later after infection.
7. The in vitro results using HIF-1a inhibition implicate the transcription factor in host cell invasion but neglect the possible impact on the epithelial cell function. Are there any differences in the production of cytokines, chemokines, or relevant antimicrobial peptides?
8. In addition, while valid, the experiments concern only very early time points when the invasion is starting and there is still no damage to the epithelial cells. It would be informative to have also data on later time points, as this might also help explain the proposed dual role of HIF-1a during infection.
9. The in vivo data is also convincing, showing that survival is improved after HIF-1a inhibition even though the fungal burden in the lungs was increased. However, the model may not exactly reflect the in vitro setting and the inhibition will inevitably affect HIF-1a in cell types other than epithelial cells, including immune cells such as neutrophils. Hence it is difficult to ascribe a specific role to HIF-1a activity in epithelial cells using this model of pharmacological inhibition.
10. To confirm the hypothesis that the benefit to survival in the mouse model might be due to improved control of inflammation, it would be essential to show a histopathological analysis of the lung tissue and also an analysis of cytokine production. Is the infection contained in the lungs or does it disseminate?
11. To what extent is the beneficial effect of HIF-1a inhibition in the susceptibility of mice to mucormycosis linked to the control of hypoxia-mediated signals during infection?

Point-by-Point Response to the Editor's and Reviewer's Comments

NCOMMS-23-27702

We thank the reviewers for their extremely helpful comments and positive feedback on our revised manuscript. Our responses are written in bold-face type.

Reviewer #1

Rhizopus delemar is an emerging opportunistic fungal pathogen often causing devastating mucormycosis in individuals with compromised immune status. Still, the mechanisms of the disease and host-pathogen interaction are not understood. An earlier study using an in vitro model of *R. delemar* and alveolar basal epithelia A549 cells suggested that the interaction between the two is vital for the disease whose process may involve the EGFR pathway of host cells. Here, Hasin and nine other colleagues re-examined the interaction using a different epithelial cell line, HSAEC1-KT, and found that *R. delemar* provokes a pro-inflammatory response. The authors found that, among several candidate host signaling pathways, the hypoxia-responsive transcription factor HIF1alpha appeared to govern the *R. delemar*-host interaction. Further studies suggested that regulation by HIF1alpha is independent of transcription and that an inhibitor of HIF1alpha, LW6, may modulate mouse survival in a murine pulmonary infection model.

Understanding disease mechanisms and host responses to fungal pathogens is critical for properly managing or treating mucormycosis infections. Therefore, the topic of the work is important. However, the work appears to be looking at a general host response that lacks any specificity. Some data showing either non-impressive or hard to interpret results suggests inadequacy in experimental approaches. While this study adds to current knowledge on this group of understudied fungal pathogens, the narrow scope of work and incremental but often conflicting findings make it unsuited to the types of audience NCOMMS is expected to attract.

Examples of issues that are of concern include:

1) "Overall lower invasion and damage of HSAEC-1 KT cells than alveolar and nasal cells": the authors did not include any of these cell lines as controls in their studies, so the conclusion may not be accurate.

We agree with this point and have removed this statement from the main text of the manuscript.

2) "No bead-beating steps in RNA extraction to reduce fungal inputs": the method may not be scientifically rigorous. RNA readings may still need to map against a fungal database(s) to eliminate those of the fungus origin, even though no database is specific to Mucorales.

We disagree that the omission of a bead-beating step indicates a lack of scientific rigor. The genomes of *Rhizopus* isolates consist mostly of genes annotated as "hypothetical proteins". Furthermore, performing gene deletion experiments with *Rhizopus spp.* is extremely difficult. Therefore, in this study, we felt it would be more informative to focus on the host side of the interaction. This choice also allowed us to save money by not generating fungus-derived sequence reads that would only sit unused on our servers. To address the concern about the fungal-derived reads, we performed a quality control step by mapping the RNA-seq reads against the *R. delemar* genome. We found that an average of 0.078% of the reads from each sample mapped to the *R. delemar* genome compared to ~95% of the reads mapping to the human genome (See Supplementary Figure 1). Additionally, the fact that we are able to

validate our results using immunoblotting, ELISAs and functional assays minimizes our concerns about our analysis being skewed by contaminating fungal reads.

3) *“These results suggest that HIF1 α governs the invasion of Mucorales into small airway epithelial cell”*: without several host signaling pathways that are potentially involved in the *R. delemar*-HSAEC-1 KT interaction and the authors examined only one such pathway with marginally significant findings, *this appears to be an overstatement.*

We do not agree that we are overstating our findings. We do agree with the sentiment, and we acknowledge, that HIF1 α is not the only host protein that governs invasion of Mucorales into airway epithelial cells. Our observation that neither LW6 nor HIF1 α siRNA completely block invasion suggests that there are other pathways/host proteins involved in this process. In fact, we have published multiple papers demonstrating the role other host proteins in invasion (Chibucos *et al.*, *Nature Communications* 2016; Watkins *et al.*, *mBio* 2018; Alqarihi *et al.* *mBio* 2020). In summary, we do think that our assertion that HIF1 α governs fungal invasion, even though it is not a complete control, is a logical conclusion based on the data.

We have included the following paragraph to the Discussion section acknowledging the possibility that other pathways also contribute to fungal invasion: “The incomplete block of invasion that we observe (Fig. 4A,C) indicates that the HIF1 α pathway is not the only pathway to govern invasion into AECs. Indeed, we have previously reported that EGFR and Integrin α 3 β 1 also govern Mucorales invasion into AECs. Further experiments are required to understand the interplay, if any, between HIF1 α , EGFR and Integrin α 3 β 1.”

4) *There are variances in RNA-Seq data between Fig 2. A and Fig 5. A, particularly in between SPRED2, IL32, and IL20, which the authors failed to discuss.*

We would like to point out that the data in Figures 1C and 4D (of the revised) manuscript represent the results from 2 independent RNA-seq experiments that were each performed in triplicate. This point has been detailed in the “Materials and Methods” section as well as in Supplementary Table 1. The minor differences in the quantity of differential expression of these genes reflect inter-experimental variation. What is important here is that we see the same upward trend. Since the samples were not normalized across both experiments, genes may shift from blue (low) to yellow (high) in one experiment and black (medium) to yellow (high) in the other experiment. Both transitions represent an increase in gene expression. We conclude that IL20 is not differentially expressed at 3 hrs in either of these experiments. Although one of the IL20 samples at 3 hours does have higher IL20 expression (yellow) in Figure 1C, the difference is not statistically significant compared to the uninfected control when all 3 samples are considered.

5) *LW6 in vivo: there is no difference in mean survival and essentially no difference in total survival ($p = 0.36$) between no treatment and 5 mg/kg treatment. Results from an increased dosage (increase of 2 days in mean survival and 40% compared to 30% at 5 mg/kg) still did not appear remarkable. Compounding the issue was that no change in fungal burden was detected. The uncertain results could only suggest that 1) the role of LW6 in governing fungal infection is marginal in addition to being indirect; 2) the experimental approach is inappropriate, which includes the authors’ methods of estimating fungal burden with qPCR and expressing the burden as log₁₀ spore/tissue weight in an infection by a filamentous fungus.*

We agree that the data with low dose LW6 is not convincing and have removed these data from the manuscript. To increase the power of the survival data following LW6 treatment, we repeated the infection experiment to obtain a total of 40 mice per experimental arm. Our

results indicate that treatment with LW6 (at a dose of 15 mg/kg) results in an increase in median survival time of 5 days with a p-value of < 0.0001.

We disagree that the experimental approach is inappropriate as we are using a very well-established animal model of mucormycosis. This model and the use of qPCR to measure fungal burden are both gold-standard methods of the field and have been established in Ashraf Ibrahim's lab.

Reviewer #2

The manuscript "Multiple roles for Hypoxia inducible factor 1-alpha in airway 1 epithelial cells during mucormycosis" by Hasin et al. explores the in vitro transcriptomic response of a cell line derived from small airway cells. This analysis reveals a significant number of differentially expressed genes (DEGs), and subsequent examination of upstream regulators identifies potential candidates involved in regulating the response of epithelial cells to Rhizopus delemar invasion. The authors specifically focus on HIF-1 α and explore its role in epithelial cell invasion, damage, and in vivo virulence. The most notable finding is the identification of the link between HIF-1 α and mucormycosis, particularly in relation to the invasion of airway epithelial cells, immunity, and virulence. The information derived from this study may hold great importance in the field and has the potential to become a landmark publication. However, there are several significant concerns that need to be addressed to firmly establish the role of HIF-1 α in mucormycosis and justify its publication in a high-impact journal such as Nature Communications.

Major concerns

1) The specificity of the transcriptomic response of epithelial cells to R. delemar is uncertain due to the absence of a control group in which cell damage is induced through alternative means. It is recommended to compare the transcriptomes of cells damaged by different methods to distinguish between the response mediated by the fungus and the response resulting from general cell damage.

Thank you for this comment. You have pointed out a significant issue that complicates all transcriptional analyses of the host response to microbial infections because infection is often very stressful to the host cells and often induces the expression of both general and specific stress response genes. To be clear, we are not making any assertions about specificity, but rather we use these RNA-seq analyses to form hypotheses that we immediately test with the use of inhibitors or siRNAs to measure contributions to infection-specific traits. This is especially important because of the possibility that a given pathway can serve two purposes (e.g., as a general stress response pathway and a microbial response pathway).

Our observation that the HIF1 α activation is contact-dependent (Figure 3D, E, F) of the revised manuscript) suggests that a specific, physical interaction triggers the increased accumulation of HIF1 α as opposed the lack of oxygen which might trigger a more general stress response.

Furthermore, our Cr51 assays indicate minimal cell damage by 6 hours post infection and ~10% - 15% damage by 12 and 24 hours, respectively (Figure 1B). Given that our transcriptional analyses were done at 3, 6, and 16 hours post-infection, the differences in gene expression that we observe are more likely a response to the infection than a general response to the cells being damaged, especially at the 3 and 6 hour timepoints.

2) Verification of the R. delemar-mediated induction of innate immune response in cultured small airway epithelial cells should be performed at the protein level.

We performed ELISAs on supernatant of AECs that have been infected with R. delemar and present data that IL1 α , IL1 β , IL8 and GM-CSF are produced in response to infections (Figure 1 of the revised manuscript). We have also performed protein-based validation of our URA

analysis by demonstrating an increase in ERK1/2 phosphorylation following infection (Figure 2 of the revised manuscript).

3) *The selection of HIF-1 α as the primary candidate involved in the response to the fungus raises concerns as several other genes identified in Figure 2B could potentially govern the response of epithelial cells to R. delemar. It should be clearly stated that multiple genes may contribute to the observed cellular response.*

We completely agree with this sentiment and we acknowledge that HIF1 α is certainly not the only host protein that governs invasion of Mucorales into airway epithelial cells. In fact, our laboratories are currently studying several pathways that we identified from this RNA-seq analysis. For the sake of focus, this current manuscript deals with our work on HIF1 α . We have included the following paragraph to the Discussion section acknowledging the possibility that other pathways also contribute to fungal invasion:

“The incomplete block of invasion that we observe (Fig. 4D) indicates that the HIF1 α pathway is not the only pathway to govern invasion into AECs. Indeed, we have previously reported that EGFR and Integrin α 3 β 1 also govern Mucorales invasion into AECs. Further experiments are required to understand the interplay, if any, between HIF1 α , EGFR and Integrin α 3 β 1.”

4) *The relevance of Figure 3A appears to be low, and it is recommended to move it to supplementary material since the results are expected based on the upstream regulatory analysis (URA) that selected HIF-1 α due to the enrichment of its targets in the transcriptomes of cells exposed to the fungus. Additionally, information regarding the datasets used in this analysis is missing.*

Thank you for this suggestion. We have now moved this heatmap to Supplementary Figure 1A.

5) *It is suggested to expand the experiments analyzing the conservation of HIF-1 α activation to include more species, considering the close phylogenetic relationship between R. delemar and R. oryzae. Simple western experiments with cultured cells can be repeated using at least Cunninghamella, Lichtheimia, Rhizomucor, and Mucor. Quantification of the Western results with a non-saturated loading control is recommended, as the signal of β -actin is saturated, making it difficult to estimate changes in HIF-1 α , particularly when the differences in the signal for HIF-1 α protein are small.*

Thank you for this comment. We have experimentally addressed this comment in two ways. First, we have now included immunoblots and accompanying graphs displaying the quantification of these blots for R. delemar, Cunninghamella bertholletiae and R. oryzae. In a complementary experiment, we performed indirect immunofluorescence on *in vitro* infections to show the nuclear localization of HIF1 α in response to infection with isolates from 5 different species of Mucorales including R. delemar, R. oryzae, Cunninghamella bertholletiae, Mucor circinelloides, and Lichtheimia corymbifera.

6) *An explanation regarding how R. delemar activates HIF-1 α is crucial to understanding the results presented in Figure 4B and 4C. These figures can be better interpreted if we consider that HIF-1 α activation occurs before spore endocytosis, explaining why the failure in activation reduces invasion. Furthermore, conducting experiments to investigate the effect of HIF-1 α upregulation on Mucorales invasion could support the hypothesis proposed by the authors. To gain further insights into the link between HIF-1 α activation and endocytosis, experiments with knockdown strains of R. delemar for*

coth7 and $\alpha 3\beta 1$ -depleted epithelial cells should be conducted, as a reduction in endocytosis is expected.

To understand how *R. delemar* activates the HIF1 α , we tested the *R. delemar*-induced accumulation of HIF1 α in HSAEC1-KT cells in the presence of Gefitinib, a potent small molecule inhibitor of EGFR. We saw no difference between Gefitinib-treated and control cells (data not shown). Since we have shown previously that the Coth7-integrin $\alpha 3\beta 1$ interaction promotes invasion by inducing EGFR signaling, this result suggests that the Coth7-integrin $\alpha 3\beta 1$ interaction does not play a role in HIF1 α pathway activation. We did notice, however, that *R. delemar*-induced accumulation of HIF1 α was almost completely abolished when the fungus and the HSAEC1-KT cells were physically separated by a Transwell membrane (Figure 3D,E, F). This result implies that physical contact is required for HIF1 α pathway activation and that the activation is hypoxia-independent.

7) Confirmation of the increase in HIF-1 α in the response of airway epithelial cells to Mucorales spore endocytosis or contact should be performed in vivo through, for example, immunohistochemical analysis.

Thank you for this comment. Using our neutropenic mouse model of mucormycosis, we performed indirect immunofluorescence analysis with an anti-HIF1 α antibody following mock-infection, infection followed by LW6-treatment, or infection followed by placebo treatment. Compared to the mock-infected samples, we see a significant increase in signal during infection (with placebo) and this signal is almost completely abolished in the lungs of LW-6 treated mice. These results are displayed in Figure 5 of the revised manuscript and text has been added to the “Results” section as well as “Materials and Methods” section.

8) A more comprehensive description of the results obtained from the RNA-seq analysis using the LW6 inhibitor is necessary to gain insights into the significance of the differential expression of 16 pro-inflammatory genes in response to LW6.

Thank you for pointing this out and we apologize for this oversight. We have addressed this in the following ways: 1) We have now included a more comprehensive description of the second RNA-seq experiment (LW6 treatment at 3 hours post-infection) into the “Results” section of the manuscript; 2) we have added the details for these 9 samples to Table S1; and 3) we have included additional supplementary tables with all of the differential expression data and the normalized read counts for these 9 datasets.

*9) The virulence assays shows only a slight reduction in virulence with the highest dose of LW6. More conclusive experiments are required, such as using mouse with tissue-specific deletion of HIF-1 α (Saini Y, Greenwood KK, Merrill C, Kim KY, Patial S, Parameswaran N, Harkema JR, LaPres JJ. Acute cobalt-induced lung injury and the role of hypoxia-inducible factor 1 α in modulating inflammation. *Toxicol Sci.* 2010 Aug;116(2):673-81. doi: 10.1093/toxsci/kfq155) and HIF-1 α upregulation (Su H, Yi J, Tsui CK, Li C, Zhu J, Li L, Zhang Q, Zhu Y, Xu J, Zhu M, Han J. HIF-1 α upregulation induces proinflammatory factors to boost host killing capacity after *Aspergillus fumigatus* exposure. *Future Microbiol.* 2023 Jan;18:27-41. doi: 10.2217/fmb-2022-0050)*

Thank you for this comment. While we agree that a tissue-specific deletion of HIF1 α would be more conclusive, these experiments are not feasible at this time due to current funding issues and the extensive amount of time it would take to perform these experiments. While we acknowledge that this work does leave certain questions unanswered, we do feel that the present work clearly supports our assertion that HIF1 α does play a role in the interaction between Mucorales both *in vitro* and *in vivo*. To increase the power of the mouse survival data

(Figure 7 in the revised manuscript), we repeated the infection experiment to obtain a total of 40 mice per experimental arm. We now observe that treatment with LW6 (at a dose of 15 mg/kg) results in an increase in median survival time of 5 days with a p-value of < 0.0001.

Minor points

1) Consider adopting the nomenclature proposed by Dolatabadi et al, 2014 (Dolatabadi S, de Hoog GS, Meis JF, Walther G. Species boundaries and nomenclature of *Rhizopus arrhizus* (syn. *R. oryzae*). *Mycoses*. 2014 Dec;57 Suppl 3:108-27. doi: 10.1111/myc.12228).

We are aware of the argument, put forth by Dolatabadi et al., that the differential abilities to produce lactic acid and fumaric acid are insufficient to support separate species designations of *R. delemar* and *R. oryzae*. However, for the sake of clarity and continuity with our previously published work on the molecular pathogenesis of *Rhizopus* isolates, we chose to use the previous nomenclature proposed by Abe et al, 2007 (Abe, A., Oda, Y., Asano, K. & Sone, T. *Rhizopus delemar* is the proper name for *Rhizopus oryzae* fumaric-malic acid producers. *Mycologia* 99, 714–722 (2007)).

2) In line 94, consider replacing “conidia” with “spores”, which is a more general term.

We have made this substitution in the text.

3) In lines 131-133, to draw this conclusion, experiments should be conducted under the same conditions, in the same laboratory, and at the same time.

We agree with this point and have removed this statement from the main text of the manuscript.

4) Supplementary Tables 2, 3, and 4 should include normalized reads for each sample.

Thank you for pointing this out and we apologize for this oversight. We have included additional supplementary tables with the normalized read counts for all of the RNA-seq datasets described in this work, including the 9 RNA-seq datasets that were inadequately presented in the original manuscript.

5) The meaning of red circles is described twice in the legend of Figure 2.

Thank you for pointing this out. We have removed the second occurrence of this sentence from the text.

6) The fungal strains used in this work should be specified.

Thank you for pointing out this oversight. We have now included text in the “Materials and Methods” section which indicates the species and strain number for each Mucorales isolate used in this study.

Reviewer #3

In this manuscript, Hasin and colleagues investigated the role of the transcription factor hypoxia-inducible factor 1 alpha (HIF-1a) during mucormycosis. The authors propose that HIF-1a plays a dual role during infection, one involving the control of host cell invasion by the fungus and the other the regulation of the immune response to infection. While this study represents an important starting point in the characterization of the role of HIF-1a in the regulation of epithelial cell function in response to

mucormycosis, it mostly describes HIF-1a induction and the effects of its inhibition on host cell invasion by the fungus, with limited mechanistic data as to why HIF-1a might have a dual role during infection, as suggested by the authors. I have several comments that may help improve the manuscript.

1) *During the invasion experiments, data is shown until 6 h after infection. Does invasion still increase at later time points, since cell damage does rises up until 48h? Also, to what extent is damage due to the fungal invasion of epithelial cells versus the presence of extracellular fungi that eventually germinate? Is there any evident fungal growth? In this regard, the authors might consider including representative microscopy images of the infected cells at relevant time points.*

The temporal dynamics of damage that we observe make it hard to determine whether the host cell damage at the later timepoints is a result of invasion or the expression of the mucorin toxin (Soliman, Nature Microbiology, 2021). Fungal growth is evident between the 6 and 16 hour timepoints such the monolayer is completely covered in a fungal mat by 16 hours post-infection. This precluded our ability to use this assay to assess fungal invasion at later timepoints. We have included representative microscopy images of the infections (Supplementary Figure 3).

2) *Regarding the RNAseq experiments, the authors mention that analysis was performed on host transcripts since no lysis of the fungal cell wall was performed and therefore no fungal RNA was available. However, an important control should be provided by showing the expression of a fungal target (e.g., 18S) and determining the relative ratio of human to fungal RNA.*

To address the concern about the fungal-derived reads, we performed a quality control step by mapping the RNA-seq seq reads, against the *R. delemar* genome. We observed that an average of 0.078% of the reads from each sample mapped to the *R. delemar* genome compared to ~95% of the reads mapping to the human genome (See Supplementary Figure 1). Therefore, we believe that the average ratio of human:fungal RNA is 1,218 : 1. This ratio and the fact that we are able to validate our results using immunoblotting, ELISAs and functional assays minimizes our concerns about our analysis being skewed by contaminating fungal reads.

3) *Given that invasion of epithelial cells is known to rely on the interaction of Coth7 in Mucorales spores with the integrin $\alpha 3\beta 1$ in host cells, the question arises as to whether this interaction is required for the activation of HIF-1a. Along the same line, is EGFR signaling required for HIF-1a induction? This is particularly relevant since, as the authors indicate, EGFR was identified as a predicted upstream regulator of HIF-1a.*

Thank you for this comment. We tested the *R. delemar*-induced accumulation of HIF1 α in HSAEC1-KT cells in the presence of Gefitinib, a potent small molecule inhibitor of EGFR, and saw no difference between treated and untreated cells (data not shown). Since we have shown previously that the Coth7-integrin $\alpha 3\beta 1$ interaction promotes invasion by inducing EGFR signaling, this result suggests that the Coth7-integrin $\alpha 3\beta 1$ interaction does not play a role in HIF1 α pathway activation.

We did notice, however, that *R. delemar*-induced accumulation of HIF1 α was almost completely abolished when the fungus and the HSAEC1-KT cells were physically separated by a Transwell membrane (Figure 3D,E, F). This result implies that physical contact is required for HIF1a pathway activation and that the activation is hypoxia-independent.

4) *Another important aspect of epithelial cell function during infection and that was not addressed – even though the authors recognize its relevance in the discussion – is the production of antimicrobial peptides and the extent to which this is regulated by HIF-1a activation in response to infection.*

Thank you for pointing this out. We went back to our datasets to specifically look for the differential expression of genes known to encode antimicrobial peptides. We found that none of them were differentially expressed following infection at any of the timepoints that we examined.

5) *The immunoblotting of HIF-1a demonstrated major differences in the cellular amounts of the transcription factor. However, while this may be indicative, together with the expression of target genes for HIF-1a, it hardly confirms the activation of HIF-1a in response to infection. Immunoblotting should instead be performed in cytosolic and nuclear extracts in order to allow the quantification of nuclear translocation of HIF-1a and therefore its activation.*

Thank you for this comment. We attempted immunoblotting on nuclear extracts but significant technical difficulties prevented us from obtaining reproducible, conclusive results. As an alternative approach to address the question of HIF1 α translocating to the nucleus during infection by Mucorales, we performed indirect immunofluorescence of HIF1 α following *in vitro* infection of HSAEC1-KT cells with 5 different species of Mucorales and in every case, we observed a significant increase in HIF1 α -derived fluorescent signal in the nucleus following infection (See Figure 3C and Supplementary Figure 2). We have included text in the “Results” section and “Materials and Methods” section to describe our findings.

6) *Along the same line, the analysis of both protein and mRNA expression should also include later time points. Given that mRNA starts to decrease at 6h, it would be important to know whether total protein also decreases later after infection.*

During the revision process, we discovered that the Figure 3B in the original manuscript was mis-labelled, indicating that the immunoblotting was performed at 3 and 6 hours post-infection when it should have read 1 and 3 hours. We apologize for this error.

We performed immunoblotting at 6 and 16 hours post-infection and did not observe a statistically significant, reproducible increase in accumulation of HIF1 α in HSEAC1-KT cells at these time points (data not shown). Our RNA-seq data shows that the HIF1 α mRNA goes down over time (Supplementary Figure 1B).

7) *The in vitro results using HIF-1a inhibition implicate the transcription factor in host cell invasion but neglect the possible impact on the epithelial cell function. Are there any differences in the production of cytokines, chemokines, or relevant antimicrobial peptides?*

The *R. delemar*-induced mRNA expression of many pro-inflammatory cytokines and chemokines are indeed sensitive to inhibition of HIF1 α (with LW6, Figure 4D,E) suggesting that HIF1 α does partially govern the ability of these airway cells to mount innate immune responses to Mucorales. We did not observe differential expression of any known antimicrobial peptides in response to *R. delemar* infection.

8) *In addition, while valid, the experiments concern only very early time points when the invasion is starting and there is still no damage to the epithelial cells. It would be informative to have also data on later time points, as this might also help explain the proposed dual role of HIF-1a during infection.*

Thank you for this insight. We agree. Unfortunately, fungal overgrowth precludes our ability to use this assay to assess fungal invasion at later timepoints.

9) The in vivo data is also convincing, showing that survival is improved after HIF-1a inhibition even though the fungal burden in the lungs was increased. However, the model may not exactly reflect the in vitro setting and the inhibition will inevitably affect HIF-1a in cell types other than epithelial cells, including immune cells such as neutrophils. Hence it is difficult to ascribe a specific role to HIF-1a activity in epithelial cells using this model of pharmacological inhibition.

Thank you for this comment. To address this, we performed indirect immunofluorescence analysis with an anti-HIF1 α antibody following mock-infection, infection followed by LW6-treatment, or infection followed by placebo treatment. Compared to the mock-infected samples, we see a significant increase in signal during infection (with placebo) and this signal is almost completely abolished in the lungs of LW-6 treated mice. These results are displayed in Figure 5 of the revised manuscript. While we agree with the reviewer that we can not ascribe a specific role of HIF1 α in epithelial cells, these results do provide evidence that HIF1 α signaling is activated during an *in vivo* lung infection.

10) To confirm the hypothesis that the benefit to survival in the mouse model might be due to improved control of inflammation, it would be essential to show a histopathological analysis of the lung tissue and also an analysis of cytokine production. Is the infection contained in the lungs or does it disseminate?

We did perform histopathologic analysis on both infected and uninfected mice but the results were inconclusive. We also observed, based on qPCR of fungal DNA, that the infection did disseminate to the brain and there was no difference between mice that were treated with LW6 or with placebo (data not shown).

11) To what extent is the beneficial effect of HIF-1a inhibition in the susceptibility of mice to mucormycosis linked to the control of hypoxia-mediated signals during infection?

The combination of our survival data following treatment with LW6 (Figure 6) and our indirect immunofluorescence staining of lungs following infection and treatment with LW6 (Figure 5) suggest that the two are indeed linked. Specifically, the mice live longer under conditions where HIF1 α protein expression is diminished by the drug.

REVIEWER COMMENTS

Reviewer #1 (Remarks to the Author):

NCOMMS-23-27702A comments

This is a resubmitted manuscript by Kavaliauskas and colleagues that examines the role of the hypoxia-inducible factor HIF1 α in experimental mucormycosis infection. The original manuscript of the same title by Hasin et al. was praised for addressing an understudied research subject, but it was lackluster in describing a general host response through often inadequate approaches and, with incremental findings, compromised the suitability of the report for NCOMMS. The authors disputed three of the five concerns raised in previous comments made by this reviewer. Still, the revision appears marginal. For example, 1) the first result reported that "fungal invasion precedes damage." It would be difficult for any readers to imagine otherwise; 2) without detecting any fungal gene expression in RNA-Seq, how could the authors convince others that host gene expression is due to the direct and specific interaction with *Rhizopus*; 3) for the potential role of LW6 on infection, the difference in mean survival remained insignificant, and there was still no difference in total survival between the test subject and the control. With neither gene names nor DNA primers listed for qPCR, the authors need to explain how qPCR-estimated spore weight can be used to measure filamentous fungal burden and how it became a "gold standard."

Reviewer #2 (Remarks to the Author):

The revised manuscript has been enhanced with the addition of experiments and missing information. Although this reviewer acknowledges the authors' efforts to strengthen the conclusions, there are some additional points that need to be addressed:

1. The results indicate that HIF1 α facilitates invasion better than what the term "governs" suggests. This term implies a major role, which is not entirely accurate. Therefore, the term "governs" should be replaced with "facilitates" or another synonym throughout the manuscript.
2. In line 220, for the sake of clarity, provide a more detailed explanation of how LW6 blocks HIF1 α . Additionally, remove repetition in lines 253-255.
3. Results with Gefitinib-treated cells should be added to the manuscript and discussed to demonstrate the independence of HIF1 α activation from EGFR, and potentially CoH7-integrin α 3 β 1 interaction. The corresponding western blots should be added as supplementary material.
4. The text in lines 255-256, "Treatment of HSAEC1-KT cells with LW6 (20 μ M) completely inhibited the *R. delemar*-induced accumulation of HIF1 α ," should be rephrased, as it may suggest a new result, despite having been presented several lines above.
5. Provide an explanation for the spherical structures observed in most overlays of infected cells in Figure 3C and Supplementary Figure 2.
6. In the legend of Figure 4B, indicate the time of exposure of cells to the siRNAs.

Reviewer #3 (Remarks to the Author):

The authors have made a substantial effort to answer the points raised, which included new data in several instances. However, I am still struggling with the authors' assumption that epithelial HIF-1 α is relevant during infection in the in vivo model. As it stands, the data does not allow concluding that. Therefore, the results in this model cannot be used to provide support to the general concept that epithelial HIF-1 α plays a role in the response to mucormycosis. Therefore, in the absence of new relevant data, this should be included as a limitation in the interpretation of the in vivo findings.

Point-by-Point Response to the Editor's and Reviewer's Comments

NCOMMS-23-27702A

We thank the reviewers for their extremely helpful comments and positive feedback on our revised manuscript. Our responses are written in bold-face type.

Reviewer #1

This is a resubmitted manuscript by Kavaliauskas and colleagues that examines the role of the hypoxia-inducible factor HIF1 α in experimental mucormycosis infection. The original manuscript of the same title by Hasin et al. was praised for addressing an understudied research subject, but it was lackluster in describing a general host response through often inadequate approaches and, with incremental findings, compromised the suitability of the report for NCOMMS. The authors disputed three of the five concerns raised in previous comments made by this reviewer. Still, the revision appears marginal.

For example, 1) the first result reported that "fungal invasion precedes damage." It would be difficult for any readers to imagine otherwise;

We feel that the idea of fungal invasion preceding damage is less obvious than Reviewer #1 is suggesting. There are many examples in the literature of microbial pathogens that can damage host cells, independently of invasion or uptake, through the use of toxins. In this case, readers would not have to imagine another possible scenario because an alternative method of damage has already been published. Specifically, Mucorales fungi produce a ricin-like toxin that is capable of killing host cells in a manner that does not necessarily first require the fungi to invade the host cells (Soliman, Nature Microbiology 2023).

2) without detecting any fungal gene expression in RNA-Seq, how could the authors convince others that host gene expression is due to the direct and specific interaction with Rhizopus;

Our inability to find many fungal reads in our RNA-seq experiments is completely expected given the fact that we did not include a bead-beating step (for the reasons detailed in our previous reply) in our isolation of total RNA from our infected samples. Inclusion of a bead-beating step, to disrupt the very rigid cell wall, is necessary for the efficient extraction of any nucleic acid from fungal cells. In other words, fungi (including Mucorales) have a cell wall and do not lyse under the relatively gentle conditions that can be used to lyse mammalian cells in tissue culture. Without cell lysis, the RNA does not get released from the cell and therefore it can not be detected.

We also object to the implication that the only way to convince others that fungi are actually in the tissue culture dish is by measuring fungal gene expression. If this were true, Reviewer #1's claims would call into question an entire literature of *in vitro* fungal infections where investigators did not explore fungal gene expression. We have provided microscopic evidence (presented in Figure 3F and in the Supplementary Material) that the fungi are indeed present.

3) for the potential role of LW6 on infection, the difference in mean survival remained insignificant, and there was still no difference in total survival between the test subject and the control.

We respectfully disagree that an increase in median survival time of 5 days (from 7 days for placebo-treated to 12 days for LW6-treated arm) and a p-value < 0.0001 is insignificant.

With neither gene names nor DNA primers listed for qPCR, the authors need to explain how qPCR-estimated spore weight can be used to measure filamentous fungal burden and how it became a "gold standard."

We apologize for this oversight and have added a citation and the primer sequences used for qPCR to the Materials and Methods section of the manuscript.

***Rhizopus spp.* are filamentous fungi. Traditional CFU counts do not accurately reflect the number of viable cells for filamentous fungi (Bowman et al., *Antimicrobial Agents and Chemotherapy*, 2001). *Rhizopus* species only grow as hyphae both in tissue and *in vitro*. When the tissue is homogenized to assess fungal burden, the long hyphae can be torn apart and often don't survive. Furthermore, those fungal (*Rhizopus*) organisms that do survive homogenization and being spread on a plate will fail to form discrete, countable colonies. Therefore, doing traditional CFU counts to assess fungal burden is wildly inaccurate. That is why we use qPCR to assess *Rhizopus* burden and express the values as genome equivalents as previously described by Ibrahim et al. *Antimicrobial Agents and Chemotherapy*, 2005, 49: p. 721–727). In this paper, it was proven that CFU counts do not correlate with progression of infection and animal death, while conidial equivalent/g of tissue as determined by qPCR did.**

Reviewer #2:

The revised manuscript has been enhanced with the addition of experiments and missing information. Although this reviewer acknowledges the authors' efforts to strengthen the conclusions, there are some additional points that need to be addressed:

1. The results indicate that HIF1 α facilitates invasion better than what the term "governs" suggests. This term implies a major role, which is not entirely accurate. Therefore, the term "governs" should be replaced with "facilitates" or another synonym throughout the manuscript.

We have replaced each of the 12 instances of 'govern' in the manuscript to 'facilitate'.

2. In line 220, for the sake of clarity, provide a more detailed explanation of how LW6 blocks HIF1 α . Additionally, remove repetition in lines 253-255.

Thank you for this comment. We have added a sentence to the manuscript to explain that LW6 inhibits the accumulation of HIF1 α by inducing the expression of the VHL protein which in turn promotes the degradation of HIF α by the proteasome. We also removed the repetition in Lines 253-255.

3. Results with Gefitinib-treated cells should be added to the manuscript and discussed to demonstrate the independence of HIF1 α activation from EGFR, and potentially CotH7-integrin α 3 β 1 interaction. The corresponding western blots should be added as supplementary material.

Thank you for this comment. We have now included an immunoblot showing that the infection-induced increase in HIF1 α protein levels are not altered by the presence of gefitinib, a potent inducer of EGFR signaling (Supplementary Figure 1C). We have also inserted text into the results section and discussion section of the manuscript.

*4. The text in lines 255-256, "Treatment of HSAEC1-KT cells with LW6 (20 μ M) completely inhibited the *R. delemar*-induced accumulation of HIF1 α ," should be rephrased, as it may suggest a new result, despite having been presented several lines above.*

Thank you for pointing this out. We agree that this statement is misleading and redundant. We have altered the text to remove this redundancy.

5. Provide an explanation for the spherical structures observed in most overlays of infected cells in Figure 3C and Supplementary Figure 2.

Thank you for pointing this out and we apologize for this oversight. The spherical structures in Figures 3C and Supplementary Figure 2 are actually fungal spores and appear because the overlay images also contain DIC images that were captured to visualize the fungal spores. These individual images were left out of the last version of the manuscript to save space. We have gone back and included the DIC images in each of the panels where we display the immunofluorescence data. This has caused us to alter the layout of Figure 3.

6. In the legend of Figure 4B, indicate the time of exposure of cells to the siRNAs.

We have indicated, in the legend of Figure 4B, that the cells were exposed to siRNAs for 48 hours prior to infection.

Reviewer #3:

The authors have made a substantial effort to answer the points raised, which included new data in several instances. However, I am still struggling with the authors' assumption that epithelial HIF-1alpha is relevant during infection in the in vivo model. As it stands, the data does not allow concluding that. Therefore, the results in this model cannot be used to provide support to the general concept that epithelial HIF-1alpha plays a role in the response to mucormycosis. Therefore, in the absence of new relevant data, this should be included as a limitation in the interpretation of the in vivo findings.

Thank you for the comment. We completely agree with this point and have added the following text to the Discussion section of the manuscript: 'We acknowledge that one limitation of our study is that the results of our *in vivo* experiments do not allow us to conclude that epithelial cell HIF1 α is important during Mucorales infection since the LW6 is likely to be inhibit HIF1 α expressed in other cell types.'

Despite this limitation, we feel that our *in vivo* data do support a novel role for HIF1 α in the pathogenesis of Mucormycosis and our *in vitro* data do provide some possible mechanistic clues as how this might. Even though we do not solve the entire puzzle, we feel that our findings do provide novel insight into the interaction between Mucorales fungi and host cells.